

# Large contrast in the vertical distribution of aerosol optical properties and radiative effects across the Indo-Gangetic Plain during SWAAMI-RAWEX campaign

Aditya Vaishya[1*], Surendran Nair Suresh Babu[1], Venugopalan Jayachandran[1], Mukunda M. Gogoi[1], Naduparambil Bharathan Lakshmi[1], Krishnaswamy Krishna Moorthy[2], Sreedharan Krishnakumari Satheesh[2,3]

[1] Space Physics Laboratory, Vikram Sarabhai Space Centre, ISRO PO, Thiruvananthapuram, India.

[2] Centre for Atmospheric and Oceanic Sciences, Indian Institute of Science, Bangalore, India.

[3] Divecha Centre for Climate Change, Indian Institute of Science, Bangalore, India.

[*]Correspondence to: Aditya Vaishya (indyaaditya@gmail.com)



## Abstract

Measurements of the vertical profiles of the optical properties (namely the extinction/ scattering and absorption coefficients; respectively $\sigma_{ext}$ / $\sigma_{scat}$ / $\sigma_{abs}$) of aerosols have been made across the Indo-Gangetic Plain (IGP) using an instrumented aircraft operated from three

base stations (Jodhpur (JDR) representing the semiarid western IGP; Varanasi (VNS)  the central IGP characterized by significant anthropogenic activities; and the industrialised coastal location in the eastern end of the IGP (Bhubaneswar, BBR)), just prior to the onset of the Indian Summer monsoon. The vertical profiles depicted region-specific absorption characteristics, while the scattering characteristics remained fairly uniform across the region,

leading to a west-east gradient in the vertical structure of single scattering albedo (SSA). Integrated from near ground to 3 km, the highest absorption coefficient and hence the lowest SSA occurred in the central IGP (Varanasi). Size distribution, inferred from the spectral variation of the scattering coefficient, showed a gradual shift from coarse particle dominance in the western IGP to strong accumulation dominance in the eastern coast with the central

IGP coming in-between, arising from a change in the aerosol type from predominantly natural (dust and sea-salt) type in the western IGP to highly anthropogenic type (industrial emissions, fossil fuel and biomass combustion) in the eastern IGP; the central IGP exhibiting a mixture of both. Aerosol induced short-wave radiative forcing, estimated using altitude resolved SSA information, revealed significant atmospheric warming in the central IGP while

a top-of-atmosphere cooling is seen, in general, in the IGP. Atmospheric heating rate profiles, estimated using altitude resolved SSA and column average SSA, revealed considerable underestimation in the latter case, emphasising the importance and necessity of having altitude resolved SSA information as against a single value for the entire column.



# 1.    Introduction

Ground based, as well as space-borne observations have established that the Indo-Gangetic plain (IGP), (the vast stretch of apparently contiguous plain land along the east-west having approximately 7 million km$^2$ area bounded between the Iranian Plateau to the west, the Bay

of Bengal to the east, the Himalayas to the north and Chotta Nagpur plateau and Aravalli ranges to the south) remains one of the aerosol hotspots in the world; depicting persistently high aerosol loading (Babu et al., 2013;Gautam et al., 2010;Dey and Di Girolamo, 2010), especially during the dry winter and pre-monsoon seasons. The increasing demographic pressure (being one of the most densely populated regions of the world), the large-scale

agricultural activities (among world's most intensely farming areas), the consequent high demand on energy (approximately 70% of the coal-fired thermal power plants of India are located in this region) and the extensive industrial activities (steel mills, cement factories, manufacturing units and a number of small and medium scale industries) are believed to be leading to consistently increasing anthropogenic emissions and hence a persistent increasing

trend in the aerosol loading as reported in Babu et al. (2013). The loose alluvial soil, which is characteristic to this region, and the semi-arid and arid regions along its western part including the Thar desert, and the prevailing complex meteorology with extreme temperatures and dry winds (except during the Indian Summer Monsoon (ISM) season) contribute their share of natural mineral aerosols. The peculiar topography of this region that

slopes down from west to east and bound on either side by high Himalayas to the north and Peninsular plateau to the South, leading to narrowing of its width from west to east aids in spatially confining and channelling these emissions until they are flushed out to the Bay of Bengal. All the above make this region a cauldron of complex aerosol types ((Moorthy et al., 2016) and references there in), which have been attracting immense scientific interest from

environmental and climate scientists, because of the known complex climate implications



(Gautam et al., 2009;Gautam et al., 2010;Lau and Kim, 2010;Lal et al., 2013;Das et al., 2015a).

Recent studies using in-situ and remote sensing methods have shown a spring-time enhancement in the aerosol optical depth and black carbon (BC) concentration in the lower free troposphere (below 5 km) over the plains and also over the Himalayas (Prijith et al., 2016;Kompalli et al., 2016;Gogoi et al., 2014), and a northward increasing gradient in the amplitude and altitude of the aerosol induced atmospheric heating Satheesh et al. (2008). In a recent study Nair et al. (2016) have found large enhancement in aerosol absorption in the lower free troposphere over the IGP during spring. Enhanced absorption by these climatically critical and highly absorbing elevated aerosols would have significant radiative implications. A very recent work, synergizing these measurements, with models and satellite data (Govardhan et al., 2017) has highlighted the potential of these elevated absorbing aerosols in aggravating stratospheric ozone loss or in delaying the recovery of ozone depletion in the past. Sarangi et al. (2016) have reported enhanced stability of lower free troposphere due to these elevated aerosols over the IGP, while Dipu et al. (2013) have found alteration in cloud water content due to these layers.

Dust aerosols are significant contributors to elevated aerosol load over the IGP during the pre-monsoon season (PMS) (Gautam et al., 2010), and along with BC constitute the major absorbing aerosol species. Desert dust aerosols, from the Arabian and Thar Desert regions, driven by winds across the IGP are found to form elevated layers of dust around 850 hpa and above (Das et al., 2013). Studies have revealed the absorbing nature of this dust (in contrast to their Saharan counterpart) (Moorthy et al., 2007), attributed to the Fe (iron) enrichment in the aerosols advected from Thar desert and adjoining semi-arid regions (Srinivas and Sarin, 2013) while modelling studies have shown a tele-connection between the advected dust and



Indian summer monsoon (ISM) (Vinoj et al., 2014). Padmakumari et al. (2013) suggested the possible role of these aerosols in acting as potential ice nuclei.

However, most of the impact assessments of aerosols over this region have used optical properties of aerosols, especially the most important parameter, the single scattering albedo

(SSA), derived either indirectly (Ramachandran et al., 2006) or from surface measurements (Ram et al., 2016), while information on the vertical structure of the optical properties (scattering, absorption, SSA) has been very sparse. This information is very important to accurately estimate the vertical structure of atmospheric heating rate resulting from absorption by aerosols. This is also necessitated by the fact that for a given amount of solar

radiation absorbed, more heating would be produced if the absorbing species is higher in the atmosphere, due to the lower density of air at higher altitudes, and trigger local convection. The knowledge of aerosol properties prior to onset of the ISM is also essential in delineating the role of aerosols as cloud condensation nuclei, and impact on cloud formation, its properties and associated precipitation. With this objective, a joint Indo-UK field campaign

South West Asian Aerosol - Monsoon Interactions (SWAAMI) has been formulated to be carried out during the onset phase of the ISM jointly with the Regional Aerosol warming experiment (RAWEX) being pursued in India under the ARFI project of ISRO's Geosphere-Biosphere Programme. One of the main aims was to characterize the vertical structure of aerosol radiative properties and estimate its impact on atmospheric thermal structure in the

IGP. For this, extensive airborne measurements of the extinction/ scattering and absorption coefficients (respectively $\sigma_{ext}$ / $\sigma_{scat}$ / $\sigma_{abs}$) were carried out across the IGP (from west to east) from three base stations, in the west, central and east IGP. The details are provided in this paper, followed by presentation of the results, and estimation of the short-wave aerosol radiative forcing and vertical profile of aerosol induced atmospheric heating rates. These

results are examined in the light of available information and the implications are discussed.



## 2.    Aircraft Campaign, Data & Methodology

### 2.1.    Campaign details

During the field experiment, the vertical structure of aerosol optical properties were measured using an instrumented aircraft (Beechcraft, B200 of the National Remote Sensing Centre (NRSC) of Indian Space Research Organisation (ISRO)) from 1st June, till 20th June, 2016; just before the onset of the ISM. The vertical profiling have been carried out from three base stations; Jodhpur (JDR), Varanasi (VNS) and Bhubaneswar (BBR), representing respectively the Western (arid), Central (Anthropogenic), and Eastern (industrialized coastal) IGP. The geographical locations of these stations are shown by the solid circles in Figure 1, which also shows the mean wind field at 850 hPa that prevailed during the campaign period. The flight tracks over each of these locations are superimposed and shown in colour-coded form JDR (green), VNS (red) and BBR (blue). Five sorties each were made (on consecutive days or in close succession) at Bhubaneswar and Varanasi, while four sorties were made from Jodhpur; the date's sorties from each station are detailed in Table 1, along with the base station details and the measurement details.  Each sortie was for ~3.5 hours, in view of the endurance of the aircraft (~ 4 hours) flying in the unpressurized mode, and comprised of measurements at 6 vertical levels (500, 1000, 1500, 2000, 2500 and 3000 m agl; above mean ground level); a typical profiling path is shown in Figure 2. After taking off from the base station, the aircraft reached the desired flight level, and after stabilizing the attitude, measurements were made to ensure a minimum duration of 25 minutes, before the aircraft climbed to the next level. For the present analysis, 5 minutes of measurements after achieving a stable level were removed as a precaution. This was done in order to avoid any spurious measurements due to sudden change in course of flight. Data points at a particular level lying outside two-sigma values of the level average were also removed. The measurements were then repeated at the new level after the aircraft has stabilised its attitude. This way 20 minutes of useful data was ensured at



each level. After measurement at the last level, the aircraft returned to the base station. Near

ground, 0 m - 200 m, data was extracted when aircraft altitude was below 200 m, as

confirmed from GPS data. Near-ground data duration was between 3 minutes to 8 minutes

each day. The measurement track had a horizontal span of ~150 km and the region of

measurement was within 300 km diameter circle centred at the base station. Details of the

flight configuration are available in earlier papers (for eg. (Babu et al., 2016;Moorthy et al.,

2004;Nair et al., 2016)).

**Table 1:** Details of the stations, dates on which flight sorties were performed and instruments operated.

| Station (Region) | Latitude ($^o$N) | Longitude ($^o$E) | Height, m (amsl) | Dates (June, 2016) | Instruments[*] |
|---|---|---|---|---|---|
| Bhubaneswar (East IGP) | 20.24 | 85.81 | 42 | 01-05 | |
| Varanasi (Central IGP) | 25.45 | 82.85 | 81 | 08, 10-13 | CAPS       PM$_{ex}$, Nephelometer, Aethalometer, |
| Jodhpur (West IGP) | 26.25 | 73.04 | 219 | 17-20 | CPC,       APS, CCNc, GPS |

\* CAPS PM$_{ex}$: Cavity Attenuated Phase Shift Extinction monitor; CPC: Condensation
Particle Counter; APS: Aerodynamic Particle Sizer; CCNc: Cloud Condensation Nuclei
counter; GPS: Global Positioning System

All the aerosol instruments aspirated ambient air through a shrouded solid diffuser inlet,

configured as detailed in Babu et al., (2016), which maintained iso-kinetic flow and the air

was supplied to the instruments through iso-kinetic flow splitters. The inlet was connected to

an external pump that maintained a volumetric flow of 70 LPM (litres per minute). More

details are available in Babu et al. (2016) and references therein.



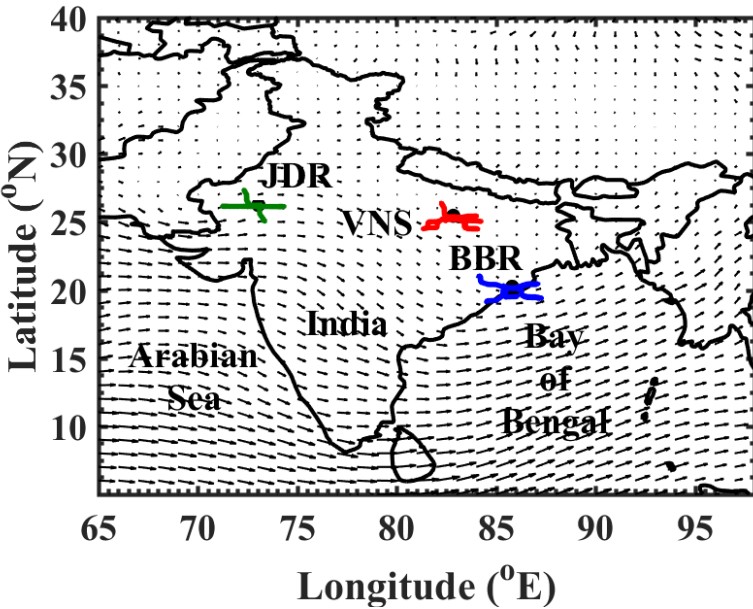

**Figure 1:** Geographical location of the aircraft campaign stations (solid circle) in the Indo Gangetic Plain superimposed on the mean wind field at 850 hPa during the campaign period. JDR, VNS & BBR stands for Jodhpur, Varanasi and Bhubaneswar, respectively. Daily flight tracks are superimposed on the stations JDR (green), VNS (red) and BBR (blue), from left to right, respectively. Each measurement track has a horizontal span of ~ 150 km from the base station.



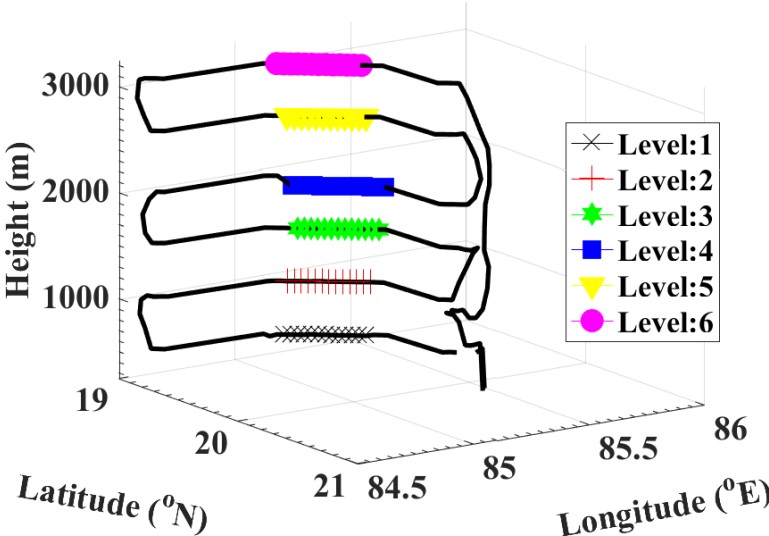

**Figure 2:** Typical course of the aircraft during a campaign sortie. Symbols represent stable levels. Each stable

level represents a minimum of 20 minutes of scientifically useful measurement.

### 2.2. Base stations

Each base station represented a distinct region of the IGP, as has already been mentioned.

Jodhpur (26.25°N, 73.04°E, 219m amsl) represented the western IGP, which stretches from

East Pakistan to northern parts of Aravalli range ending in Delhi, is characteristically arid

region, dominated by natural aerosols (mineral dust). It also contains the 'Great Indian

Desert' or Thar Desert. Consequently, during summer the temperature often exceeds 40°C

during daytime; with the maximum values reaching as high as 48 °C. Pre-monsoon aerosol

system over this region is dominated by dust, primarily produced locally and that transported

from Arabia, the Middle-East and eastern Africa (Prasad and Singh, 2007).

Central IGP, extending from north-eastern boundaries of the Aravalli range up to north-west

regions of the Chota Nagpur plateau, is represented by Varanasi (25.45°N, 82.85°E, 81 m

amsl). Central IGP hosts numerous coal fired thermal power plants, large scale industries



including steel and cement factories, and has highest population density as compared to other regions of the IGP. Approximately ~ 65% of the area in the central IGP is under cultivation. Central IGP is frequented by local dust storms and transported dust (Prasad and Singh, 2007) during the pre-monsoon season.

East IGP, geographically bound by Chotta Nagpur Plateau in the west, Himalayas in the north, Purvanchal hills in the east, and Bay of Bengal in the south, is represented by Bhubaneswar (20.24°N 85.81°E, 42m amsl), located about 70 km inland of the coast. It encompasses a large swath of land with numerous water bodies, dense forested regions, and the great Sundarbans delta. Apart from local emissions from industries, vehicles and other

household practices, East IGP receives a significant portion of its aerosol load from the west and central IGP (Nair et al., 2007), due to its location in the continental outflow from the central IGP. Bhubaneswar and adjoining regions are hosts to several heavy industries and thermal power plants, and as such, high aerosol optical depth (~0.5 at 500 nm) prevails over this region (Das et al., 2009) .

**2.3.    Instruments, measurements and database**
A suit of instruments has been used aboard for measuring the aerosol properties (see Table 1), of which the data from those dealing with the optical properties are used in this study. These included, aerosol light extinction coefficient ($\sigma_{ext}$) measurements at 530 nm, carried using a Cavity Attenuated Phase Shift Extinction Monitor (CAPS PM$_{ex}$) (Model PM$_{ex}$ of Aerodyne

Research Inc.); Aerosol light scattering coefficient ($\sigma_{scat}$) measured using a three wavelength (450, 500 and 700 nm) integrating Nephelometer (TSI; Model: 3563); and aerosol absorption coefficient ($\sigma_{abs}$) derived from the measurements made using a 7-channel Aethalometer (model AE-33 of Magee Scientific). The CAPS PM$_{ex}$ employs the cavity attenuated phase shift technology (Herbelin and McKay, 1981;Kebabian et al., 2007) and measures the phase

shift in the light leaving a highly reflective optical cell illuminated by a square wave





modulated light emitting diode source (Massoli et al., 2010). $\sigma_{ext}$ is calculated from the differences of phase shift between the particle-free air and particle-laden air in the optical chamber. Details are given by Massoli et al., (2010). This instrument was operated at a flow rate of 0.85 LPM. Auto baseline measurements were performed every 2 minutes. Massoli et

al., (2010) have established that CPAS $PM_{ex}$ has a detection limit of 3 $Mm^{-1}$ or less at 1 second time resolution.

The details of the Nephelometer operation and principle of measurement are given by (Anderson et al., 1996;Heintzenberg and Charlson, 1996) . The instrument was operated at a flow rate of 16 LPM, and calibrated with $CO_2$ span gas before and after the campaign to

ascertain consistency in performance. Besides this, zero background measurement with filtered air was done on hourly basis to ascertain the health of the instrument. The measurements are corrected for the well-known truncation error (due to non-availability of measurements for angles $<7^0$ and $> 170^0$ following Anderson and Ogren (1998) methodology as detailed in earlier papers (Nair et al., 2009;Babu et al., 2012).

The Aethalometer provides aerosol BC mass concentration ($M_{BC}$) during each measurement. Aerosol light absorption coefficient ($\sigma_{abs}$) was then calculated from $M_{BC}$ using Equation 1.

$$\sigma_{abs} = M_{BC} * m_{BC} \tag{1}$$

where $m_{BC}$ is the mass absorption cross-section of BC (= 7.77 $m^2.g^{-1}$).  The Aethalometer was operated at a flow rate of 2 LPM and data frequency was set to 1 minute. Measurements by

the Aethalometer are known to have the instrument artefacts viz. multiple scattering, loading effect and assumption of $m_{BC}$ (Weingartner et al., 2003;Liousse et al., 1993). The under-estimation of BC due to loading effect is compensated in the instrument, which uses the dual-spot technique, following Drinovec et al. (2015).  A factor of 1.57 is used to compensate for the enhanced light absorption arising due to multiple scattering within the filter fibre matrix

(Drinovec et al., 2015). Details of Aethalometer data analysis can also be found in earlier publications (Babu and Moorthy, 2002;Moorthy et al., 2004). Aethalometer data was corrected for volumetric flow, in order to sample same volume of air at each altitude, following Moorthy et al. (2004).

All on-board computers and instruments were time-synchronized with global positioning system (GPS) time during each sortie. After each sortie, the measured data were geo-referenced using high time resolution (1s) GPS data, available from a GPS receiver aboard.

## 3.    Results and Discussion

### 3.1.    Vertical & spatial distribution of aerosol radiative properties

The raw data of $\sigma_{ext}$, $\sigma_{scat}$ and $\sigma_{abs}$, after all necessary corrections and time-tagging, from all the sorties at a particular station, have been grouped in terms of the different altitude levels chosen for the sortie (as described in section 2.4) and averaged to construct the mean, station- specific altitude profile. All the three parameters, $\sigma_{ext}$, $\sigma_{scat}$, and $\sigma_{abs}$ are presented for 530 nm wavelengths (the wavelength used by the CAPS), and for this the $\sigma_{scat}$ and $\sigma_{abs}$ values

were interpolated (between at 450 nm and 550 nm for $\sigma_{scat}$ ; and between 520 nm and 590 nm for $\sigma_{abs}$) using the corresponding Ångström power law relation (Ångström, 1964) :

$$\sigma_{scat/abs} = \beta_{scat/abs} \cdot \lambda^{\alpha_{scat/abs}} \qquad (2)$$

where $\beta_{scat/abs}$ is a constant, $\lambda$ is wavelength and $\alpha_{scat/abs}$ is scattering/absorption Ångström exponent.

Figures 3 a, b, and c show the vertical distributions, respectively of $\sigma_{ext}$, $\sigma_{scat}$ and $\sigma_{abs}$, over the three stations. In all the figures, symbols square, triangle, and circle correspond to



measurements over JDR, VNS, and BBR, respectively. Error bars represent the standard error at that level for the station.

A vertical heterogeneity is clearly seen in all the properties across the IGP. While the altitude variation is very weak at JDR (western IGP) and moderate at BBR, it is rather strong at the

central IGP (VNS). The weak vertical variation at JDR is attributed to the strong convective mixing over this arid region, where the solar heating is very intense during this season. Above around 1.5 km, there is a decrease $\sigma_{ext}$ and $\sigma_{scat}$, which is stronger than that seen in $\sigma_{abs}$. This is likely to be due to rapid sedimentation of heavier dust particles, which contribute largely to $\sigma_{ext}$ and $\sigma_{scat}$, as compared to their anthropogenic counterpart which contribute

dominantly to $\sigma_{abs}$. The extinction at 3 km is just half of that at 0.5 km or even at 1.5 km. The day-to-day variability over the western IGP is smaller compared to that at the other two regions as evidenced by the shorter error bars. This is also attributed to the near-uniform dominance of dust aerosols in this region and the scarcity of anthropogenic sources of aerosols e.g. industries, coal fired power plants etc. In contrast, VNS in the central IGP shows

significantly higher values of $\sigma_{ext}$, $\sigma_{scat}$, and $\sigma_{abs}$ close to the surface (clearly attributed to the large abundance of anthropogenic emissions in this region, as has been stated earlier) and a rather sharp decrease with altitude, with $\sigma_{ext}$ at 3 km falling by a factor of 4 of the near surface value (similar for $\sigma_{scat}$ and $\sigma_{abs}$). As the Central IGP is dotted with numerous coal fired power plants (http://www.ntpc.co.in), heavy industries, and has highest population

density in the range (800-1200 $km^{-2}$) the resulting large emissions are reflected in the high values and the large day-to-day variability (large standard error bars) of the optical properties of aerosols over this region. While close to the surface, the extinction values are considerably higher over Central and Eastern IGP (compared to the western part) at the higher levels (above 2 km) the values are of comparable magnitudes at all the three stations, showing a

larger spatial homogeneity in the lower free troposphere. Interestingly, the absorption





coefficient over BBR in the east IGP remains nearly steady with altitude up to around 2 km, above which it increases (unlike at the other two stations) showing more absorbing aerosols aloft.

Spatially, the column averaged values of $\sigma_{scat}$, $\sigma_{abs}$ and $\sigma_{ext}$ (up to the maximum height up to

5   which measurements were made) are the highest in the central IGP compared to the eastern and western IGP; primarily due to the very high values near the surface. Station average values of $\sigma_{ext}$, $\sigma_{scat}$ and $\sigma_{abs}$ along with standard error are given in Table 2.

**Table 2:** Mean ± standard error of column averaged (from near ground to 3000 m) aerosol radiative properties.

| Parameter | Specific regions over the IGP | | |
|---|---|---|---|
| | West (JDR) | Central (VNS) | East (BBR) |
| $\sigma_{ext}$ (Mm$^{-1}$) | 79 ± 6 | 95 ± 19 | 75 ± 12 |
| $\sigma_{scat}$ (Mm$^{-1}$) | 63 ± 5 | 69 ± 14 | 58 ± 6 |
| $\sigma_{abs}$ (Mm$^{-1}$) | 16 ± 2 | 26 ± 9 | 15 ± 3 |
| $\alpha_{scat}$ | 0.9 ± 0.2 | 1.7 ± 0.2 | 2.0 ± 0.1 |
| SSA (integrated) | 0.84 ± 0.03 | 0.73 ± 0.06 | 0.79 ± 0.06 |





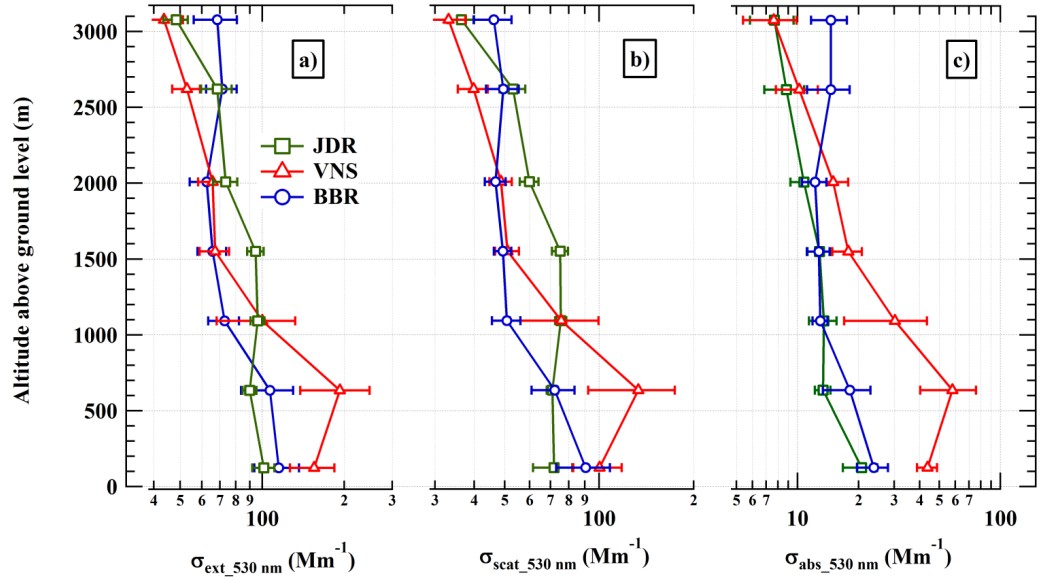



**Figure 3:** Mean altitudinal variation of a) $\sigma_{ext}$, b) $\sigma_{scat}$, and c) $\sigma_{abs}$ over JDR (square), VNS (triangle) and BBR (circle) in logarithmic scale. Error bars represent the corresponding standard errors (standard deviation of the mean).

Aerosol scattering / absorption Ångström exponents ($\alpha_{sca/abs}$) were derived from the respective coefficients as a function of wavelength using the relation,

$$\alpha_{scat/abs} = -\frac{ln\left(\frac{\sigma_{scat/abs}(\lambda_1)}{\sigma_{scat/abs}(\lambda_2)}\right)}{ln\left(\frac{\lambda_1}{\lambda_2}\right)} \qquad (3)$$

$\alpha_{abs}$ values are indicators of potential aerosol types. $\alpha_{abs}$ value ~1 indicates absorbing aerosols mainly from fossil fuel sources (Kirchstetter et al., 2004;Russell et al., 2010) whereas values

> 2 are indicative of absorbing aerosols from biomass sources and dust (Russell et al., 2010;Weinzierl et al., 2011). Range of $\alpha_{abs}$ values decreased gradually from desert regions of the west IGP, 1.1 – 2.4, through anthropogenic aerosols dominated central IGP, 1.1 – 1.6, to coastal east IGP, 1.1 – 1.5. Detailed analysis on this aspect will be presented in subsequent works. $\alpha_{scat}$ gives an indication of the dominant particle size mode (Schuster et al., 2006).

Higher $\alpha_{scat}$ value suggests increased sub-micron dominance and vice-versa. In the present case, we calculated $\alpha_{scat}$ using $\sigma_{scat}$ at wavelengths 450 nm and 700 nm. $\alpha_{scat}$ values < 0.4 indicate a super-micron mode aerosol dominance (Smirnov et al., 2002) and values > 2 suggests sub-micron aerosol dominance (Schuster et al., 2006), while values in between 0.4 and 2 are due to a mix of aerosol distribution with varying degree of super-micron and sub-

micron particles. Figure 4 shows the altitudinal variation of $\alpha_{scat}$ for the three regions. It clearly reveals the gradual transformation of aerosol type across the IGP during the pre-monsoon. The optical properties over Western IGP is mostly controlled by the coarse mode (dust) aerosols, leading to remarkably low values of $\alpha_{scat}$ in the entire altitude regime (values



lying in the range 0.7 to 1.0 with extremely weak altitude variation), compared to the other two regions. On the other hand, despite of its coastal proximity, the industrialized eastern IGP (BBR) has the highest value of $\alpha_{scat}$ values, remaining well above 2.0 for the altitude region below 2 km, with a weak decrease above. The weak decrease observed may be due to the

presence of long-range advection of dust aerosols across the IGP, at higher altitudes. Such long distance travel resulted in gravitational settling of coarser dust leaving behind fine accumulation mode aerosols. The central IGP, with its strong anthropogenic emissions adding to the advected dust, reveals a transition from the coarse mode dominant western IGP to the fine mode dominant eastern IGP; with $\alpha_{scat}$ lying in the range 1.4 to 1.9 depicting a

conspicuous decrease with increase in altitude. While the higher values of $\alpha_{scat}$ closer to the surface signifies the strong contribution of anthropogenic emissions from the region leading to significant increase in the submicron aerosol loading (as also evidenced by the large values of scattering and absorption coefficients in Figure 3), the increase in fractional contribution of advected dust results in a decrease on $\alpha_{scat}$ of the composite aerosols at higher altitudes.

The column averaged values of $\alpha_{scat}$ (Table 2) increases from ~0.9 in the western IGP through 1.7 over the central IGP to 2.0 over the eastern IGP. In summary, as we move from west to east in the IGP the aerosol type changes from super-micron mode dominant natural aerosols (dessert dust) to sub-micron mode dominant anthropogenic aerosols (mix of aerosols from industrial sources, fossil fuel and biomass burning.



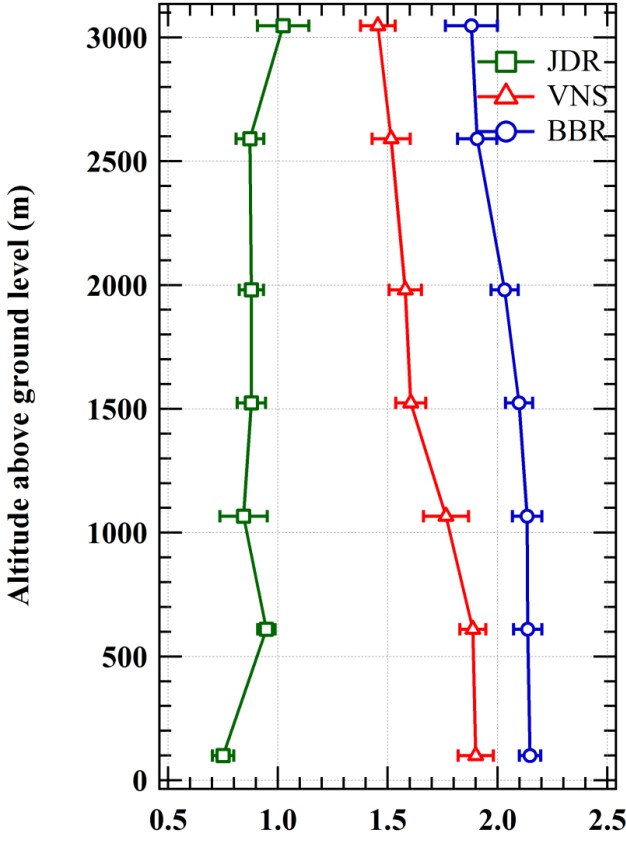

**Figure 4:** Altitudinal variation of α for the aircraft campaign stations: JDR (square), VNS (triangle), and BBR

(circle). Error bars represent the standard error in the mean of level averaged α.

5    SSA at 530 nm has been estimated $(SSA(\lambda) = \dfrac{\sigma_{scat}(\lambda)}{\sigma_{scat}(\lambda) + \sigma_{abs(\lambda)}})$ from the

concurrently measured spectral $\sigma_{scat}$ and $\sigma_{abs}$, the coefficients interpolated to 530 nm using

Ångström power law relation described in Equation 2. SSA values were then layer averaged

over each location and the mean altitude profiles are shown in Figure 5, which shows the

strikingly differing absorption properties of aerosols over different regions of IGP as well as



distinctly differing vertical variation. The highest SSA and lowest absorption occurs over the arid regions of western IGP (JDR), dominated by natural mineral dust aerosols. The SSA values here are well above 0.8, with the exception of near ground SSA values which dipped to a low of ~0.78, and little or very weak altitude variation, though a very weak decrease is

indicated above 2.5 km. The column integrated SSA in the western IGP is 0.84. In sharp contrast, the least value of SSA (with a column integrated value of 0.73) occurs over the central IGP region (VNS), indicating strong aerosol absorption. The SSA values go below 0.75 (as low as 0.7 at 500 m altitude) closer to the surface and within the boundary layer (below 1.5 km) and increases steadily with altitude and reach values close to those over JDR

when the altitude reached ~3 km. Interestingly, over the industrialized eastern IGP represented by BBR, where fine submicron aerosols dominated throughout the altitude profiled, the aerosols are only moderately absorbing. Close to the surface moderately high SSA values are seen (0.8), decreasing marginally with altitude to reach 0.79 at 2 km. Above this, SSA decreases more rapidly, indicating increased aerosol absorption in the lower free

troposphere, and the values drop to 0.76 by the time altitude increased to 3 km (and is the lowest among the three regions at this altitude). Above about 2.5 km, strongest aerosol absorption occurs in the eastern IGP, even though the column integrated value (0.79) lies between that seen for the western and Central IGP and this will have implications in layer-heating by aerosol absorption.

SSA values found in the present study for the western IGP are lower than those reported by Verma el al. (2013), ~ 0.89, from AERONET retrievals, where the SSA values were for severe dust storm episodes when coarse mode scattering dust is dominant. For the central IGP, Ram et al. (2016) have reported SSA value of ~0.77 prior to onset of the ISM which is in line with the finding of the present study. Quite opposite to central IGP, East IGP shows

reduction in SSA values with enhanced variability as we move to higher altitudes.



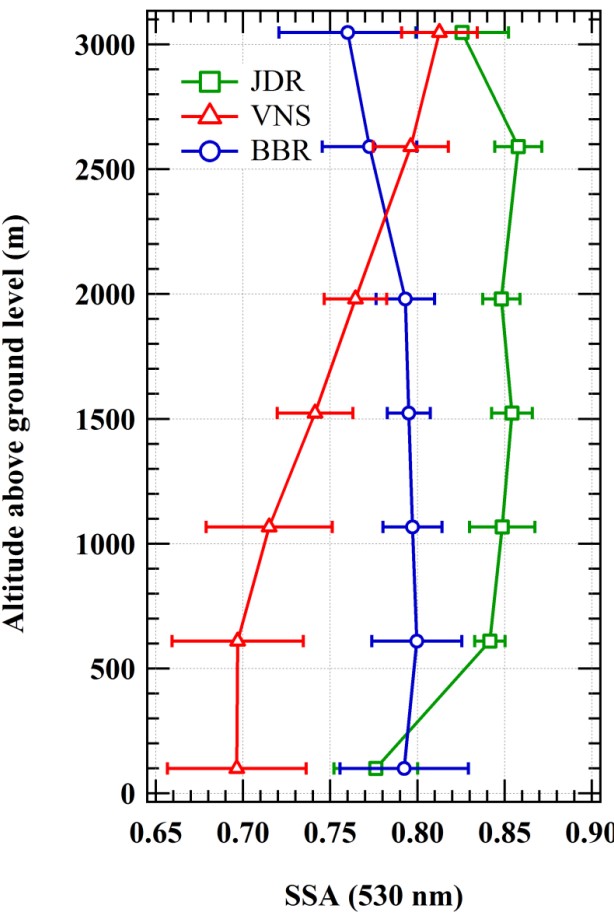

**Figure 5:** Altitudinal variation of SSA at 530 nm for the aircraft campaign stations: JDR (square), VNS (triangle), and BBR (circle). Error bars represent the standard errors.

This clearly points to the inaccuracies that may arise in the estimation of aerosol absorption

5 and atmospheric forcing and heating rates using a single columnar SSA value (derived from, for example sunphotometer measurements) and brings out the need for region-specific, altitude resolved values of SSA and estimate layer-by-layer forcing and heating rates for more accurate climate impact assessment. This is addressed in a subsequent section.

Based on airborne measurements during the winter of 2012 and spring of 2013 Babu et al.

10 (2016) have reported altitudinal profiles of SSA and its seasonality over central Indian



regions. They found an enhancement in aerosol absorption in the free troposphere during spring over the IGP in general. Combining our results with those reported by Babu et al. (2016), and assuming the inter-annual variations to be less significant, we present in Figure 6 the temporal evolution of aerosol absorption (integrated over the altitude region up to about

5    3.5 km above mean sea level) over the IGP from winter through spring to just prior to the onset of the ISM. While a significant reduction in SSA, indicating increase in aerosol absorption, occurs from winter to pre-monsoon; more strong is the change from spring to 'prior to onset of the ISM' period (i.e. from March-April to June). This may be due to lower horizontal ventilation in the IGP due to wind field reversal prior to onset of the ISM, thus

10   leading to build-up of absorbing aerosols (Vaishya et al., 2017).


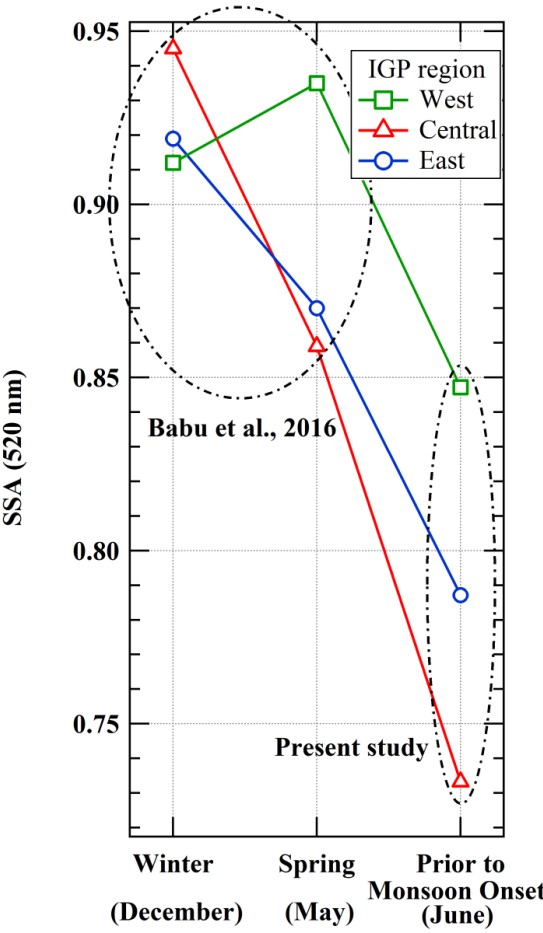

**Figure 6:** Seasonal variation of columnar (500 m - 3000 m) SSA over different regions of the IGP: west (square), central (triangle), and east (circle). Values in the upper left circle are taken from Babu et al., 2016.

Our values are compared with values of SSA reported over different regions of Indian

5   mainland and adjoining region, as reported by different investigators, in Table 3, along with

the methodologies/techniques used to compute it is indicated by symbols, explanation for

which is given at the footer of the table.

**Table 3:** SSA values reported over the Indian landmass and adjoining oceans.

| Region | Location | Period/ Season | Wavelength (nm) | SSA | Surface/ Columnar | References |
|---|---|---|---|---|---|---|
| Himalayas | Nainital | Winter | 550 | 0.93 | Surface[&] | (Dumka and |



| Region | City | Season/Period | λ (nm) | SSA | Type | Reference |
|---|---|---|---|---|---|---|
| | | | | | | Kaskaoutis, 2014) |
| | | Dec 2005/ Winter | 500 | 0.9 | Columnar@ | (Pant et al., 2006) |
| West IGP | Dehradun | Winter Spring | 520 | 0.931 0.904 | Columnar% (surface to 3000 m) | (Babu et al., 2016) |
| | Jodhpur | Winter Spring | | 0.891 0.935 | | |
| | | Prior to Monsoon onset | 530 | 0.84 | Columnar%4 (near surface to 3000 m) | **Present study** |
| | Hisar | Dec 2004/ Winter | 500 | 0.88 | Columnar@ | (Ramachandran et al., 2006) |
| | Jaipur | Winter Spring | 520 | 0.904 0.910 | Columnar% (surface to 3000 m) | (Babu et al., 2016) |
| Central IGP | Delhi | Winter Spring | 550 | 0.74 0.63 | Surface% | (Soni et al., 2010) |
| | Kanpur | Winter Spring | 678 | 0.81 0.76 | Surface* | (Ram et al., 2016) |
| | Lucknow | Winter Spring | 520 | 0.878 0.859 | Columnar% (surface to 3000 m) | (Babu et al., 2016) |
| | Varanasi | Prior to Monsoon onset | 530 | 0.73 | Columnar% (near surface to 3000 m) | **Present study** |
| East IGP | Gandhi College | Spring | 675 | 0.89 | Columnar& | (Srivastava et al., 2011) |
| | Ranchi Patna | Winter Spring | 520 | 0.927 0.870 | Columnar% (surface to 3000 m) | (Babu et al., 2016) |
| | Bhubaneswar | Prior to Monsoon onset | 530 | 0.79 | Columnar% (near surface to 3000 m) | **Present study** |
| | Kolkata | Spring | 500 | 0.8-0.85 | Columnar& | (Talukdar et al., 2017) |
| North-East | Dibrugarh | All seasons | 500 | 0.8 | Columnar@ | (Pathak et al., 2010) |
| Central India | Nagpur | Winter Spring | 520 | 0.889 0.790 | Columnar% (surface to 3000 m) | (Babu et al., 2016) |
| Peninsular India | Hyderabad | Winter | 550 | 0.83 | Columnar@ | (Sinha et al., 2012) |
| | | Spring | 520 | 0.880 | Columnar% (surface to 3000 m) | (Babu et al., 2016) |
| | Chennai | Feb – Mar, 2001 | 500 | 0.77 | Surface@ | (Ramachandran, 2005) |
| | Bengaluru | Oct – Dec, 2001 | 500 | 0.73 | Surface@ | (Babu et al., 2002) |
| | Thiruvananthapuram | Winter Spring | 500 | 0.77 0.8 | Columnar@ | (Babu et al., 2007) |



| Arabian Sea | Arabian Sea | March, 1999 | 530 | 0.9 | Surface[%] | (Jayaraman et al., 2001) |
|---|---|---|---|---|---|---|
| | | Dec 2008-Jan 2009/ Winter | 550 | 0.88 | Surface[%] | (Babu et al., 2012) |
| Bay of Bengal (BoB) | Entire BoB | Mar-Apr, 2006/ Spring | 550 | 0.93 | Surface[%] | (Nair et al., 2008) |
| | | Apr-May, 2006/ Spring | 550 | 0.94-0.98 | Surface[%] | (Moorthy et al., 2009) |
| Indian Ocean (IO) | North IO | Feb – Mar, 1998 | 534 | 0.9 | Surface[%] | (Satheesh et al., 1999) |
| | | March, 2006 | 400-700 | 0.89 | Column[&] | (Ramana et al., 2007) |

*: derived from attenuation measurement of filter samples; &: retrieved from sun/sky radiometer; %: retrieved from in-situ absorption and scattering; @: using a combination of in-situ aerosol optical depth and constrained OPAC model output.

One consistent feature that emerges from Table 3 is that there is a significant reduction in the SSA (or increase in aerosol absorption) over the Himalayan foothills, the IGP regions, and Central India in spring and towards pre-monsoon season, as compared to the winter season. This is not the case with peninsular India and adjoining oceanic regions where spring time SSA values show an increase indicating less absorbing aerosol in the atmosphere. This

apparent dichotomous behaviour of SSA, from north to south, need further investigation, in terms of the possible role of transported dust. Satheesh et al. (2008) have reported an increase in aerosol induced heating rates from northern Indian Ocean to Central India concomitant with an increase in $\sigma_{ext}$, at 3 km altitude. This, they concluded was due to presence of elevated aerosols at increasingly higher altitudes as one moves from northern Indian Ocean to

Central India. SSA from simultaneous and direct measurement of $\sigma_{scat}$ and $\sigma_{abs}$ is more accurate when compared to those retrieved from sun photometer sky radiance measurements or estimated from modelling studies. Most of the SSA values reported for the Indian region




are retrieved indirectly and hence susceptible to respective inversion or model uncertainties. In this context, the results from present study assume great significance.

## 3.2. Dust fraction over the IGP

In order to delineate the possible role of long-range transported dust to the observed vertical

heterogeneity over the IGP regions, cloud free vertical profiles of $\sigma_{ext}$ at 532 nm from the Space-borne lidar CALIOP (The Cloud-Aerosol Lidar with orthogonal Polarization) aboard CALIPSO (The Cloud-Aerosol Lidar and Infrared Pathfinder Satellite Observations) satellite were examined. Cloudy profiles were screened out based on Liu et al. (2010) using Cloud Aerosol Discrimination (CAD) score in the range -70 to -100. Dust extinction coefficient was

estimated from CALIPSO level-2 depolarization measurements over a period of one month from 20th May 2016 up to 20th June 2016, overlapping with aircraft campaign measurement period, over a 2$^o$ x 2$^o$ spatial grid centred at the campaign base stations. Reported values of depolarization ratio of dust aerosols are in the range 0.2 - 0.3 while non-dust/spherical aerosols have much lower depolarization ratio of magnitude in the range 0.02 - 0.07 [Yu ,

2015]. Based on this criterion, dust extinction coefficient ($\sigma_d$) is obtained from the CALIOP measurements of PDR ($\delta_p$), total back-scattering coefficient($\beta$), and with the a priori information on the typical values of depolarization ratio of dust ($\delta_d$) /non-dust particles ($\delta_{nd}$) and the lidar ratio of dust aerosols ($S_d$) following Tesche et al., (2009). This method is advantageous in separating the pure dust extinction coefficient in mixed systems. CALIOP

profiles of aerosol extinction coefficient averaged during the measurement period were normalized using the mean MODIS (Collection 6 - MYD08_D3_6_AOD_550_Dark_Target_Deep_Blue_Combined_Mean) AOD during the same period over the study regions (Levy et al., 2013).

Figure 7 shows the mean altitudinal profiles of aerosol extinction coefficient (dash line) and

dust extinction coefficient (dot line), derived from CALIPSO as mentioned above. The



profiles are smoothed by performing a 3-point running average. Dust fraction, or the contribution of dust extinction to total extinction, is also shown (solid line). The highest dust fraction (dust contribution to the total extinction) occurs over the western IGP (JDR), where it shows a steady increase with altitude from~10 - 20% at 300 m to close to 100% above 2

5      km, vindicating the earlier inference drawn from the spectral variation of $\alpha_{scat}$. As we move to central IGP, the dust seems to be well mixed through the column (with a small peak at ~ 1.5 km). Moreover, it should also be kept in mind that while the dust over the western IGP is pristine in nature, that over the central IGP is more absorbing in nature because of its mixing with other anthropogenic emissions (such as BC) (Srivastava et al., 2012), as also has been

10     indicated by high $\alpha_{scat}$ values in the range 1.4 - 1.9. The least dust fraction (< 10%) is observed over BBR (eastern IGP), which is far away from dust source regions and has $\alpha_{scat}$ values ~ 2.0, and has significant industrial and anthropogenic emissions.





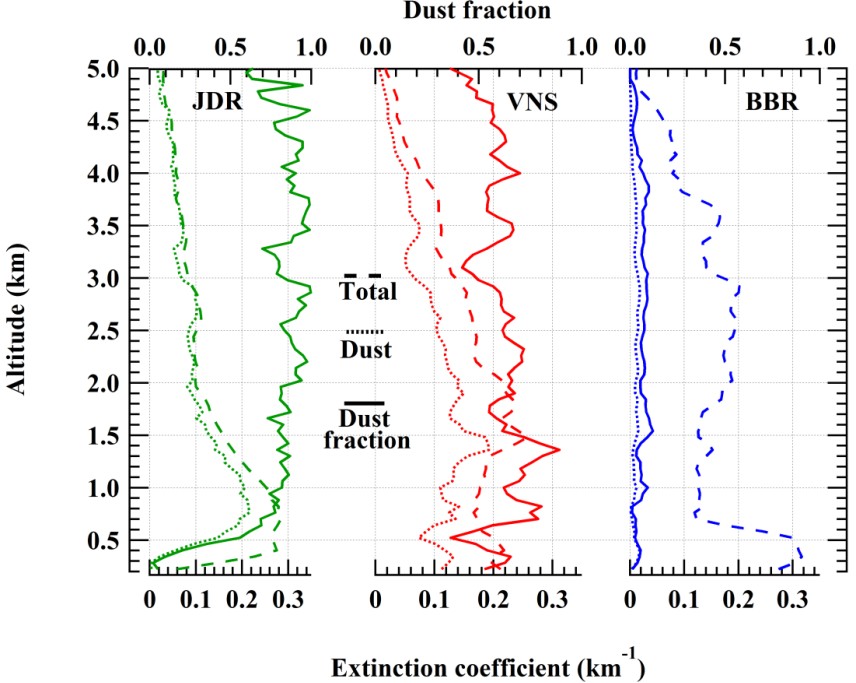

**Figure 7:** CALIPSO derived total (dash line) and dust (dot line) extinction profiles over the stations JDR (left), VNS (centre) and BBR (right). Also shown is the dust fraction (solid line) for the three stations.

### 3.3.    Aerosol radiative forcing and atmospheric heating

5    A Discrete Ordinate Radiative Transfer (DISORT) based model (Santa Barbara DISORT Atmospheric Radiative Transfer (SBDART) (Ricchiazzi et al., 1998)) was used to estimate aerosol forcing on shortwave fluxes. Inputs to the model were latitude, longitude, day of year, surface pressure at the station, surface albedo, and spectral values of AOD, and SSA, and Legendre moments of the aerosol phase function. Spectral surface albedo values for all the

10    stations were taken from MODIS Albedo product (Level 3; MCD43C) (Schaaf et al., 2002). Spectral values of AOD for each level were obtained from the sum total of layer integrated values of in-situ spectral $\sigma_{scat}$ and $\sigma_{abs}$ assuming a well-mixed layer of 200 m above and below the measurement altitude. SSA values for each level were obtained from the ratio of spectral $\sigma_{scat}$ to sum of spectral $\sigma_{scat}$ and $\sigma_{abs}$. Legendre moments of the aerosol phase function





were calculated using the asymmetry parameter (g) which was obtained using the Henyey-Greenstein approximation (Wiscombe and Grams, 1976). SBDART model simulation for short-wave radiative flux estimation was performed with and without aerosols with eight radiation streams at an interval of 1 hour and then diurnally averaged. The net short-wave

aerosol radiative forcing (ARF) at the top-of-atmosphere (TOA) and surface (SUR) was then

computed as $\Delta F_{TOA/SUR} = Flux_{\_withaerosol\ TOA/SUR} - Flux_{\_withoutaerosol\ TOA/SUR}$. Difference between TOA and surface forcing is the atmospheric forcing represented as $\Delta F_{ATM} = \Delta F_{TOA} - \Delta F_{SUR}$.

Figure 8 shows the aerosol induced short-wave radiative forcing ($\Delta F$) at the top-of-atmosphere (TOA) – filled black bar, at surface (SUR) – filled white bar, and in the

atmosphere (ATM) – filled gray bar, for the three regions of the IGP: west (JDR), central (VNS), and east (BBR). The net effect of aerosols on the $(\Delta F)_{TOA}$ and $(\Delta F)_{SUR}$ is cooling, in W m$^{-2}$,  - 8.2 & - 16.9 in the west IGP,  - 6.5 & - 22.9 in the central IGP, and  - 5.4 & -15.3 in the east IGP, respectively, with decreasing magnitudes of the TOA forcing from west to east, while a sharp peak in the central IGP at the surface. The atmospheric forcing, $(\Delta F)_{ATM}$ (W m$^{-2}$),

which represents the amount of radiation absorbed or trapped by the atmosphere and thus results in heating of the atmospheric layer, is maximum in the central IGP (16.4) followed by east and west IGP, where the $(\Delta F)_{ATM}$ values are 9.9 and 8.7, respectively.





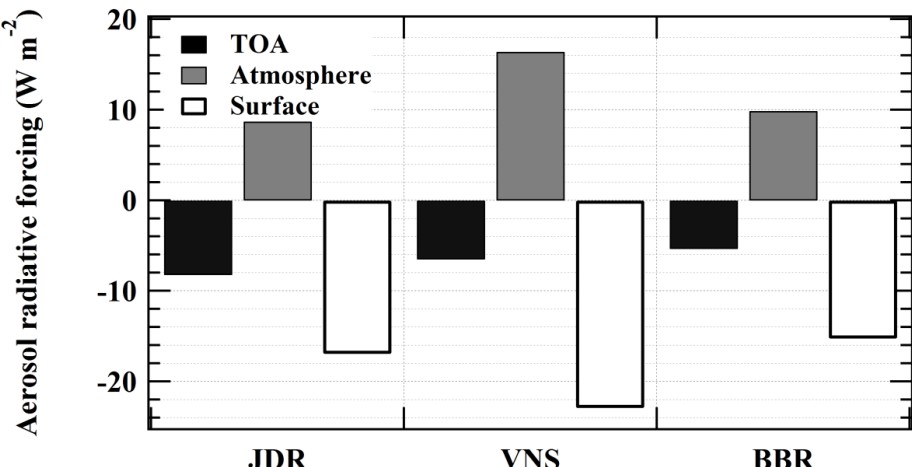

**Figure 8:** Aerosol induced short-wave radiative forcing at the top-of-atmosphere (TOA) (filled black bar), surface (filled white bar), and atmosphere (filled gray bar) for the regions of the IGP: west (JDR) - left; centre (VNS) - centre; and east (BBR) - right panel.

The forcing values found in the present study are examined against those reported from other

stations in the IGP, as listed in Table 4. The $(\Delta F)_{TOA}$ for Jodhpur is comparable with Jaipur

for the pre-monsoon season but the $(\Delta F)_{ATM}$ and $(\Delta F)_{SUR}$ are significantly different. $\Delta F$ for

Varanasi and Kanpur, both in the central IGP, are in good agreement for pre-monsoon

10    seasons. For Bhubaneswar and Kolkata, both in the east IGP, there are significant differences

in $(\Delta F)_{TOA}$, $(\Delta F)_{ATM}$ and $(\Delta F)_{SUR}$ values. While $(\Delta F)_{TOA}$ is negative for Bhubaneswar its

positive for Kolkata, but for the winter season, indicating a net gain of energy by the Earth-

Atmosphere system. The differences in the $\Delta F$ estimates are likely due to change in surface

type, variation in column abundance of aerosols and their vertical distribution, SSA etc.



**Table 4:** Aerosol induced short-wave radiative forcing at various locations in the IGP.

| Location in the IGP | Season | Aerosol induced short-wave radiative forcing ($\Delta F$) (W m$^{-2}$) | | | References |
|---|---|---|---|---|---|
| | | TOA | ATM | SUR | |
| Jaipur | Pre-monsoon | -8.2 | 21.4 | -29.5 | (Verma et al., 2017) |
| | Monsoon | -9.1 | 17.2 | -26.3 | |
| | Post-monsoon | -9.8 | 17.4 | -27.1 | |
| | Winter | -24.7 | 17.3 | -7.5 | |
| Jodhpur | Prior to monsoon onset | -8.2 | 8.7 | -16.9 | **Present study** |
| Hisar | Winter | -3 | 18 | -21 | (Ramachandran et al., 2006) |
| Kanpur | Winter | -14.5 | 34.6 | -49.1 | (Kaskaoutis et al., 2013) |
| | Pre-monsoon | -12.8 | 44.2 | -57.0 | |
| | Monsoon | -17.1 | 25.4 | -42.5 | |
| | Post-monsoon | -17.6 | 29.5 | -47.0 | |
| Varanasi | Prior to monsoon onset | -6.5 | 16.4 | -22.9 | **Present study** |
| Bhubaneswar | Prior to monsoon onset | -5.4 | 9.9 | -15.3 | **Present study** |
| Kolkata | Winter | 5.1 | 75.4 | -70.3 | (Das et al., 2015b) |

The impact of radiation absorbed by an aerosol layer in the atmosphere is represented in terms of atmospheric heating rate (Liou, 2002). Aerosol induced heating rate was calculated

5    using Equation 5:

$$\frac{\partial T}{\partial t} = \frac{g}{C_p} \frac{\Delta F}{\Delta P} \qquad (5)$$



where $\partial T/\partial t$ is the heating rate (K d$^{-1}$), g is the acceleration due to gravity, $C_p$ is the specific heat capacity of air at constant pressure and $\Delta P$ is the atmospheric pressure difference between top and bottom boundary of the layer, and $\Delta F$ is aerosol induced forcing in the layer.

Figure 9 shows altitudinal profile of the heating rates, thus estimated, over each of the regions. While the solid line in the figure represents heating rate profiles calculated using in-situ level averaged values of aerosol properties, the dash-dot line represent identical calculation but with single columnar SSA value for each station. Heating rate profiles for west, central, and east IGP regions are distinctively different. Western IGP shows a gradual increase in heating as we move away from the surface, attains maxima at 1.5 km and then reduces as we go up. Central IGP has maximum heating near to surface and it decreases with altitude. East IGP exhibits an increase in heating with altitude. Near surface (500 m - 1000 m) heating maxima in west IGP (JDR) is due to enhanced absorption by dust aerosols which are present in significant amount near to surface (JDR - Figure 7), as evident from high extinction coefficient and high dust fraction. In central IGP (VNS) significantly absorbing aerosol layer, with SSA ~ 0.7, is present from surface to ~ 1 km. This combined with enhanced aerosol loading at 500 m layer, results in high absorption and thereby significant heating of the layer. Over the eastern IGP (BBR) coarse mode dust subsides due to gravitational setting but fine mode dust mixed/coated with anthropogenic aerosols may still be present in appreciable amount at higher altitudes. This results in significant layer heating at ~ 3000 m over BBR where dust fraction increases from ~ 0% at surface to ~10% at 3 km.

Comparing the heating rate profiles obtained using altitude-resolved SSA with those estimated using the columnar SSA in Figure 9 emphasises the importance of the knowledge of the altitude profile. The use of single SSA value for the entire column overestimates the heating at higher levels in western IGP and largely underestimates, by as much as 0.2 K.day$^{-}$



[1], in the eastern IGP. In the central IGP it underestimates the heating rates at lower altitudes, specifically at layers with significant absorbing aerosol loading.

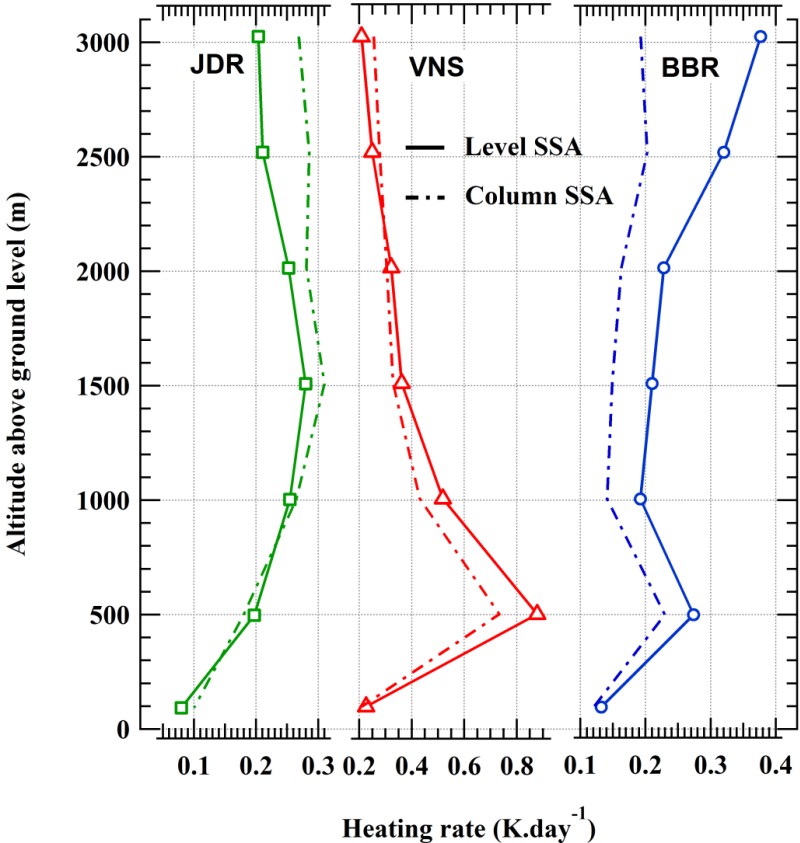

**Figure 9:** Altitudinal heating rate profiles for the stations i) JDR (left), ii) VNS (center), and BBR (right). Solid lines are for heating profiles calculated using in-situ aircraft profile data and layer averaged aerosol properties. Dash dot line represent heating profiles calculated using in-situ aircraft profile data and column averaged aerosol SSA.

Contrary to the findings of Kuhlmann and Quaas (2010), who found a peak in heating rate in the IGP at around ~2.5 km, our study shows that over central IGP maximum heating occurs around 0.5 km and in east IGP it is around 2.5 - 3.0 km, bringing out a spatial variation across the IGP. Significant reduction in SSA values at higher altitudes in East IGP may results in



enhanced atmospheric heating and surface dimming. This might reduce the temperature gradient between surface and atmosphere leading to reduced boundary layer convection (Feng et al., 2016). Analysis of a decadal (2001-2010) aerosol data, from AERONET station, revealed mean and peak heating rates in the range 0.6 - 1.1 K d$^{-1}$ and 0.9 - 1.7 K d$^{-1}$,

respectively, in Kanpur (Kaskaoutis et al., 2013) in the central IGP. Sarangi et al., [2016] have shown that due to enhanced absorption by aerosols at altitudes 1.5 km and above there is reduction in incoming solar flux in lower troposphere and associated cooling of about 2 to 3 $^{o}$C in the IGP. The horizontal and vertical gradients in the heating rate found in the present study depict the complex nature of influence of aerosols on atmospheric stability over the

Indian landmass. A similar scenario exists over the adjoining oceanic regions, Arabian sea and Bay of Bengal as well; where a gradient in aerosol-induced atmospheric heating rate was found which increased from ~0.1 K d$^{-1}$ in the south-western Arabian Sea to as high as ~0.5 K d$^{-1}$ over the north-eastern Bay of Bengal (Nair et al., 2013).

Radiative and hydrological implications of the observed decrease in SSA over the entire IGP,

prior to onset of the ISM, needs detailed investigation using numerical models. Gradient in vertical heating rates, both regionally and longitudinally in the IGP, might induce anomalous radiative effects leading to reduced land-atmosphere thermal contrast (Feng et al., 2016) and enhanced stratification of lower troposphere (Barbaro et al., 2013;Babu et al., 2002). This in a cumulative way can alter regional precipitation patterns (Chung and Zhang, 2004). Ignoring

the vertically resolved SSA, and instead using column average values, may lead to considerable underestimation/overestimation of aerosol induced heating rates for atmospheric layers with highly absorbing/scattering aerosols, respectively. The significant findings, detailed in earlier sections, have implications to atmospheric stability, associated circulation patterns, and possible modulations to onset of the ISM and regional precipitation.



## 4. Conclusions

An aircraft campaign was conducted, from $1^{st}$ - $20^{th}$ June, 2016, to characterize aerosol radiative properties, both extrinsic and intrinsic, and its impact on atmospheric thermal structure prior to the onset of the Indian Summer Monsoon (ISM). The three base stations, Jodhpur (JDR), Varanasi (VNS) and Bhubaneswar (BBR) were aptly selected to represent west, central and east IGP, respectively. Exhaustive measurements of aerosol light scattering and light absorption properties were carried out to quantify enhanced absorption by aerosols. Major findings from the study are as following:

1. As we move from west to east in the IGP the aerosol type changes from super-micron mode dominant natural aerosols (dessert dust), $\alpha_{scat} \sim 0.9$, to sub-micron mode dominant anthropogenic aerosols (mix of aerosols from industrial sources, fossil fuel and biomass burning etc.), $\alpha_{scat} \sim 2.0$.

2. Central and east IGP have opposite SSA trends. While SSA in central IGP increases vertically that in the east IGP decreases. This significant reduction in SSA values, from ~0.8 to ~0.76, at higher altitudes in the east IGP may results in enhanced atmospheric heating and surface dimming.

3. The heterogeneous altitudinal SSA point towards inaccuracies that may arise in the estimation of aerosol absorption and atmospheric forcing and heating rates using a single columnar SSA value and highlights the need for region-specific, altitude resolved values of SSA and estimate layer-by-layer forcing and heating rates for more accurate climate impact assessment.

4. Aerosols across the Indo-Gangetic Plain become highly absorbing prior to onset of the Indian Summer Monsoon, compared to winter and spring.



5. Aerosol induced short-wave radiative forcing at the top-of-atmosphere, surface and atmosphere were -8.2, -16.9, 8.7 W.m$^{-2}$ for west IGP (JDR), -6.5, -22.9, 16.4 W.m$^{-2}$ for central IGP (VNS), and -5.36, -15.3, 9.9 W.m$^{-2}$ for east IGP (BBR), respectively.

6. Atmospheric heating rate profiles with layer resolved SSA and column averaged SSA values differed significantly for highly absorbing/scattering aerosol layers. While usage of column average SSA underestimated the heating for highly absorbing aerosol layers it did opposite for scattering aerosol layers.

To sum up, a system of highly absorbing aerosols, with SSA values as low as ~0.7, prevails over the IGP prior to onset of the ISM. This, combined with the fact that elevated absorbing aerosol layers may occurs at different altitudes across the IGP, has implications to atmospheric stability, time of monsoon onset and regional precipitation patterns. Further studies, combining aerosols radiative properties with cloud parameters viz. cloud optical depth, cloud albedo and fraction etc. will help in discerning the effects of enhanced absorption prior to onset of the ISM on regional climate.

## Competing interests

The authors declare that they have no conflict of interest.

## Acknowledgements

This study was carried out as part of the SWAAMI-RAWEX campaigns. We thank Director, National Remote Sensing Centre (NRSC), Hyderabad and the Aerial Services and Digital Mapping Area (AS & DMA) for providing the aircraft support for this experiment. The authors wish to thank the crew of the aircraft for their help throughout the field campaign and the wholehearted support of the NRSC aircraft team. Aditya Vaishya is supported by the Department of Science and Technology, Government of India, through its INSPIRE Faculty Programme. MERRA data was obtained from website http://mirador.gsfc.nasa.gov/. MODIS albedo products were obtained from NASA's Earth data portal. CALIPSO extinction profiles are obtained from http://www-calipso.larc.nasa.gov/tools/data_avail. Details of ARFINET data and aircraft data used in this manuscript and the point of contact are available at http://spl.gov.in; "Research Themes;" "Aerosol Radiative Forcing Section." The SWAAMI project was supported by MoES.



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
