# Peer review of "© Author(s) 2018. CC BY 4.0 License."

_Atmospheric Chemistry and Physics, 2018_

## Referee Comment (RC1) · Anonymous Referee #1 · 19 Aug 2018

The paper reports aircraft measurements of aerosol optical properties collected in three locations in the Indo-Gangetic plain, just prior to the onset of the Indian summer monsoon. The authors report also on the results of radiative forcing and heating rates calculations. They conclude that the knowledge of the vertical distribution of the aerosol optical properties is key to perform accurate calculations. The dataset is surely interesting, and the paper is overall clearly written. I believe this work can be important to the community and should be published after addressing a few issues that I will discuss next.

[Figure]

GENERAL COMMENTS The main issue with this analysis is a lack of careful discussion of the uncertainties associated with the different quantities presented. Such uncertainties can affect how the data are interpreted and especially how strong the conclusions might be. Therefore, before publication, the authors should discuss and include an analysis of the uncertainties to account for random as well as systematic errors (for example, the corrections applied to the aethalometer data are mentioned, but uncertainties associated with those methods and parameters are not included nor discussed). Some specific cases are discussed in the next section. Although, of minor importance from a scientific point of view, the authors should consider revising the use of the articles "the" and "a" in the manuscript, sometimes misused, but especially, often missing. Also, the use of some preposition and the punctuation could be improved.

SPECIFIC COMMENTS Page 6, lines 23-34: "Data points at a particular level lying outside two-sigma values of the level average were also removed." The rationale behind this data filtering choice needs to be provided. It is not clear how such data manipulation might bias the averages. Page 6, lines 25: "...aircraft has..." maybe should be "...aircraft had..." Page 6, lines 25: comma after "This way"? Page 7, lines 2: "...data was extracted when..." what do the authors mean by "extracted"? Do they mean removed? Or something else? Please clarify. Page 11, lines 18: What wavelength was used and why this specific mass absorption cross-section? Please clarify. Even more importantly, and related to the general comment above, what is the uncertainty associated with the use of this specific mac? There is a wide range of mac values published in the literature, even for a specific given wavelength and even for laboratory generated aerosols; therefore, this certainly introduces a significant uncertainty in the aerosol absorption and SSA, and will propagate into derived forcings and heating rates. Page 11, lines 23: The choice of the factor value of 1.57 needs some discussion. In addition, certainly, some rather large uncertainty is associated with this parameter; that should be discussed and should be accounted for while discussing the significance of differences between different layers (these values could easily change even in the same region) and especially when comparing with other results published in the liter-

ature that might have used different correction factors, for example. Page 12, lines 7: "aboard" maybe should be "onboard"? Page 14, line 2: Even just from the statistical error bars presented in the graph (meaning even ignoring the other potential sources of error discussed above), the increase in absorption coefficient seems hardly significant in a statistical sense. Table 2: As mentioned, in the table, the authors should add other uncertainties (e.g., instrument accuracies and precisions etc...) Page 16, lines 8-11: Specific to the interpretation of the aethalometer Ångsröm exponent, the authors might be interested in the paper by Fialho, P., Freitas, M. C., Barata, F., Vieira, B., Hansen, a. D. a., & Honrath, R. E. (2006). The Aethalometer calibration and determination of iron concentration in dust aerosols. Journal of Aerosol Science, 37(11), 1497–1506. https://doi.org/10.1016/j.jaerosci.2006.03.002 Page 16, line 21: "... reveals the gradual transformation of aerosol..." the term "transformation" might be misinterpreted as the aging of an aerosol, but I believe the authors mean something different. Consider using a different term or clarifying. Page 19, line 25: Is the "reduction" statistically significant? See similar comments above. Page 20, line 4: "This clearly points..." The conclusion seems quite reasonable here, but I am not sure the data strongly ("clearly") support this point because of the large standard errors that make several of the changes hardly significant. Maybe using terminology such as "...suggest" or "...data are compatible with..." might be more appropriate. Figure 6 requires error bars. Page 24, line 13: "This, they concluded was..." consider adding a comma before was. Figure 7 would greatly benefit from error bands. Page 28, line 9: The acronyms TOA and SUR were defined already just a couple of lines above. Figure 8 requires error bars. Page 31, line 21-23: Such comparison needs an uncertainty estimate and the discussion needs to be put in the context of the uncertainties. See previous comments. Figure 9 requires error bars. Page 33, line 3: Consider revising the sentence "Analysis of a decadal (2001-2010) aerosol data..." to "Analysis of decadal (2001-2010) aerosol data..." or "Analysis of a decadal (2001-2010) aerosol dataset...". In the same sentence, "AERONET station" is this a specific station? If so, please provide the location. Or are they several "stations" Page 33, line 6: "of" in front of "1.5 km"? Also, consider

adding a comma after "above" and before "there". Page 33, line 8: Are these "2 to 3 C" or "2 to 3 C/d"? Also, consider using consistent units (meaning K instead of C) Page 34, line 3: ". . .both extrinsic and intrinsic. . ." what does that mean? Do the authors mean ". . .both intensive and extensive. . ." or something else? Page 34, line 14: The term "significant" should probably be used exclusively with a statistical meaning; so, is this "reduction" indeed significant from a statistical point of view? Or not? See also comments above. Page 34, line 22: the verb "become" seems to imply a transformation of the aerosol over time, while here I think the different sources and transport patterns might be more dominant? Page 35, line 5: see the previous comment on the use of "significantly" Page 35, line 10: "occurs" should be "occur"; also, ". . ..implications to. . ." or ". . .implications for. . ."?

---

## Referee Comment (RC2) · V. R. Kotamarthi (Referee) · 15 Sep 2018

The manuscript describes aerosol optical property measurements using a small aircraft platform during pre-monsoon period over Northern India during 2016. This is likely that first such reported measurements from this part of India and hence quite interesting. In large part these measurements establish previous reports on aerosol property variations going from West to east over this part of India and provides some insights into the variability in altitude. A key missing component in the discussion is an evaluation of mixing and its affect of the vertical profiles reported here. Unfortunately,

the aircraft is limited in its ability to reach higher latitudes (non-pressurized) and hence limited to 3000m above ASL. I suspect this may be within the PBL for all the measurements reported here and the profile is primarily established by strong emissions during the sampling time and possibly an artifact of the time the measurements were made. A slightly later time would probably erase this vertical variability. The PBL height at all three locations is probably above 3km during the pre-monsoon time. I would like to see some more discussion of the meteorological/dynamic state of the atmosphere to put the measurements in context. a)what time of the day where these measurements made? Where all the measurements at all these sites made at the same time of the day? b) what is the PBL height at the time of measurements? Are there any measurements of temperature profiles that are available close to the regions where the aircraft flew? c) Manoharan et al., 2013 established that the aerosol absorption in this region has a correlation to the size of the particle. Is there any data on aerosol size that can be shared? d) measurements over eastern India indicate less absorbing particles (may be more sulfate?). It will be helpful to have a more extended discussion the scattering at the eastern site and how it may relate to radiative forcing. I agree with the 'cauldron of complex aerosol type's'. (page 3, line 24) and it may be time to start focusing on the 'other' aerosols in the region besides just absorbing aerosols (sulfates and nitrates). These and SOA may have significant health impacts for the region and deserve more attention.

---

## Author Comment (AC1) · 2 Nov 2018

We thank both the reviewers for their constructive comments which have helped us to improve the manuscript. Below are the reviewer comments, in plain italics, and authors replies, in bold.

**acp-2018-686-RC1**

The paper reports aircraft measurements of aerosol optical properties collected in three locations in the Indo-Gangetic plain, just prior to the onset of the Indian summer monsoon. The authors report also on the results of radiative forcing and heating rates calculations. They conclude that the knowledge of the vertical distribution of the aerosol optical properties is key to perform accurate calculations. The dataset is surely interesting, and the paper is overall clearly written. I believe this work can be important to the community and should be published after addressing a few issues that I will discuss next.

**GENERAL COMMENTS**

The main issue with this analysis is a lack of careful discussion of the uncertainties associated with the different quantities presented. Such uncertainties can affect how the data are interpreted and especially how strong the conclusions might be. Therefore, before publication, the authors should discuss and include an analysis of the uncertainties to account for random as well as systematic errors (for example, the corrections applied to the Aethalometer data are mentioned, but uncertainties associated with those methods and parameters are not included nor discussed).

We thank the reviewer for constructive comments. In the revised manuscript we have discussed the uncertainties associated with various parameters, and have interpreted the results in light of those.

Uncertainties in  $\sigma_{scat}$ :

Major uncertainty associated with the scattering measurements is due to non-ideality of the Nephelometer viz. not able to sense the scattering signal for angles  $<7^{0}$  and  $> 170^{0}$  – referred to as truncation error. Scattering measurements were corrected for the truncation error following Anderson and Ogren (1998) and Anderson et al. (1996). They have estimated uncertainties in  $\sigma_{scat}$  measurements to be within ~ ±10%.

Following lines have been added to the manuscript:

Pg 11, Ln14-15: Uncertainties in measured  $\sigma_{scat}$  are within ~ ±10% (Anderson et al., 1996).

Uncertainties in  $\sigma_{ext}$ :

Pg 11, Ln 4-6: Massoli et al., (2010) have established that the CAPS  $PM_{ex}$  has a detection limit of 3 Mm-1 or less at 1 second time resolution and has an uncertainty of ±3%.

Uncertainties in  $\sigma_{abs}$ :

Pg 12, Ln 4-16: Uncertainties related to measurement of absorption coefficient, using filter based techniques, have been discussed in a series of literatures (Müller et al., 2011;Drinovec et al., 2015;Collaud Coen et al., 2010;Segura et al., 2014;Lack et al., 2014). These uncertainties mainly stem from two major reasons: i) multiple-scattering within the filter fiber matrix, and ii) lower attenuation coefficients for higher filter loadings – filter loading effect (Weingartner et al., 2003). Lack et al. (2014) have estimated an uncertainty of 12 % - 30 % in  $\sigma_{abs}$  measured using filter-based techniques. However, this assumption is on the higher side for the present study for two reasons: i) the new generation Aethalometer (Drinovec et al., 2015) has in place real time compensation of loading effect which earlier was assumed as a constant; and ii) usage of advanced filter tape material which minimizes the effect due to multiple scattering and can be better characterized. After taking into consideration uncertainties introduced due to flow instabilities (Drinovec et al., 2015) and an uncertainty of ~ 10% is expected in the absorption coefficient measurements.

Uncertainties in  $\alpha_{scat/abs}$ :

Pg 17, Ln 8-9: Based on the uncertainties in the measurement of  $\sigma_{scat}$  and  $\sigma_{abs}$ , as described in section 2.3 above, the estimated uncertainty in  $\alpha_{scat/abs} \sim 14\%$ .

Uncertainties in SSA :

Pg 19, Ln 7-8: 12%-13% uncertainty is estimated in SSA.

Some specific cases are discussed in the next section. Although, of minor importance from a scientific point of view, the authors should consider revising the use of the articles "the" and "a" in the manuscript, sometimes misused, but especially, often missing. Also, the use of some preposition and the punctuation could be improved.

Complied with. The manuscript has been revised as per the reviewers suggestion and the usage of articles 'the' and 'a' have been appropriately revised.

**SPECIFIC COMMENTS**

Page 6, lines 23-34: "Data points at a particular level lying outside two-sigma values of the level average were also removed." The rationale behind this data filtering choice needs to be provided. It is not clear how such data manipulation might bias the averages.

**Complied with.**

It was found that due to occasional appearance of clouds aerosol number concentration increased from otherwise stable values. In order to remove such unavoidable incidences from influencing aerosol properties two-sigma criteria was applied. Overall, < ~ 3% of the measurements were screened out due to this criteria. A uniform criterion was applied to screen in all the measurements.

Page 6, lines 25: "....aircraft has...." maybe should be "....aircraft had...."

**Complied with.**

Page 6, lines 25: comma after "This way"?

**Complied with.**

Page 7, lines 2: "....data was extracted when...." what do the authors mean by "extracted"? Do they mean removed? Or something else? Please clarify.

Sorry for the confusion. The sentence is modified as 'Near ground data represent measurements during the ascend/descend phase of the aircraft between ground and 200 m AGL'.

Page 11, lines 18: What wavelength was used and why this specific mass absorption crosssection? Please clarify. Even more importantly, and related to the general comment above, what is the uncertainty associated with the use of this specific mac? There is a wide range of mac values published in the literature, even for a specific given wavelength and even for laboratory generated aerosols; therefore, this certainly introduces a significant uncertainty in the aerosol absorption and SSA, and will propagate into derived forcings and heating rates.

Sorry for the confusion here. Aethalometer measures attenuation of light by aerosols deposited on a filter spot. Absorption coefficient is then calculated from the rate of change of attenuation, filter spot area, and volumetric flow rate using equation given below (Weingartner et al., 2003).

$$\sigma_{abs=\frac{A}{Q}*\frac{\Delta ATN}{\Delta t}}$$

where A is the filter spot area, Q is the volumetric flow rate, and  $\Delta ATN$  is change in attenuation in time  $\Delta t$ .

BC at 880 nm wavelength is calculated from  $\sigma_{abs}$  by assuming a mass-absorption crosssection (mac) value of 7.77 m2.g-1. This value denotes the mac of freshly emitted BC aerosols freely suspended in the Air. Hence there is no uncertainty involved in  $\sigma_{abs}$  due to an assumption of mac. The only uncertainty which will propagate in the calculation of SSA, radiative forcing, and heating rates is the uncertainty in the measurement of absorption coefficient itself. As discussed earlier, estimation of absorption coefficient using filter based optical attenuation technique incurs two major uncertainties: one associated with multiple scattering, and the other with loading effect (Weingartner et al., 2003;Coen et al., 2009). The dual spot Aethalometer (Drinovec et al., 2015) used in the present study compensates in real time for loading effect. Based on a 2-week Klagenfurt campaign data Drinovec et al. (2015) have calculated the value of the parameter C accounting for multiple scattering correction. They concluded that the value of C, obtained as 1.57 for TFE coated glass filter used in AE33, is strongly dependent on filter tape material used.

Following changes have been made in the manuscript:

Pg 11, Ln 16-21: The Aethalometer measures attenuation of light by aerosols deposited on a filter spot. Absorption coefficient is then calculated from the rate of change of attenuation, filter spot area, and volumetric flow rate using equation given below (Weingartner et al., 2003).

$$\sigma_{abs=\frac{A}{Q}*\frac{\Delta ATN}{\Delta t}}$$

**where A is the filter spot area, Q is the volumetric flow rate, and $\Delta ATN$ is change in attenuation in time $\Delta t$ .**

Page11, lines 23: The choice of the factor value of 1.57 needs some discussion. In addition, certainly, some rather large uncertainty is associated with this parameter; that should be discussed and should be accounted for while discussing the significance of differences between different layers (these values could easily change even in the same region) and especially when comparing with other results published in the literature that might have used different correction factors, for example.

The factor C, with value 1.57 in the present case, represents the enhancement in optical absorption due to filter tape and varies depending on the filter tape material and its packing density. For the Aethalometer model - AE-33 filter tape used is Tetrafluoroethylene (TFE) - coated glass filter (Pallflex 'Fibre-film'- T60A20) and has a C value of 1.57 (Drinovec et al., 2015). Value of C is constant for a given filter type.

Page 12, lines 7: "aboard" maybe should be "onboard"?

**Complied with.**

Page 14, line 2: Even just from the statistical error bars presented in the graph (meaning even ignoring the other potential sources of error discussed above), the increase in absorption coefficient seems hardly significant in a statistical sense.

We agree with the reviewer. The sentence has been modified now and it reads 'Absorption coefficient over BBR in the east IGP remains nearly steady with altitude up to around 2 km, above which it marginally increases (unlike at the other two stations), albeit the increase is within the natural variability of lower levels, showing more absorbing aerosols aloft.'

*Table 2: As mentioned, in the table, the authors should add other uncertainties (e.g., instrument accuracies and precisions etc...)*

**As suggested in the earlier comments, uncertainties related to various parameters have been explained in the text hence not mentioned here to avoid repetition.**

Page 16, lines 8-11:Specific to the interpretation of the aethalometer Ångsröm exponent, the authors might be interested in the paper by Fialho, P., Freitas, M. C., Barata, F., Vieira, B., Hansen,a. D. a., &Honrath, R. E. (2006). The Aethalometer calibration and determination of iron concentration in dust aerosols. Journal of Aerosol Science, 37(11), 1497–1506.https://doi.org/10.1016/j.jaerosci.2006.03.002

**Thanks for bringing to our attention this important literature which certainly will be useful while interpreting absorption Ångsröm exponent.**

Page 16, line 21: ".... reveals the gradual transformation of aerosol...." the term "transformation" might be misinterpreted as the aging of an aerosol, but I believe the authors mean something different. Consider using a different term or clarifying.

Complied with. The sentence has been modified. It now reads as: 'It clearly reveals a gradual change in aerosol type across the IGP during the pre-monsoon.'

Page 19, line 25: Is the "reduction" statistically significant? See similar comments above.

**The sentence has been modified:**

Pg 21, Ln 1: 'Quite opposite to central IGP, East IGP shows reduction in SSA values, albeit within the uncertainty range, with enhanced variability as we move to higher altitudes.'

Page 20, line 4: "This clearly points...." The conclusion seems quite reasonable here, but I am not sure the data strongly ("clearly") support this point because of the large standard errors that make several of the changes hardly significant. Maybe using terminology such as "....suggest" or "....data are compatible with...." might be more appropriate.

Complied with. The sentence has been modified and now reads as: 'This suggest that inaccuracies may arise in the estimation of aerosol absorption and atmospheric forcing and heating rates when using a single columnar SSA value (derived from, for example sunphotometer measurements) and brings out the need for region-specific, altitude resolved values of SSA and estimate layer-by-layer forcing and heating rates for more accurate climate impact assessment.'

Figure 6 requires error bars.

Complied with.

**Figure 6:** Seasonal variation of columnar (500 m - 3000 m) SSA over different regions of the IGP: west (square), central (triangle), and east (circle). Values in the upper left circle are calculated from Babu et al., 2016.

Page 24, line13: "This, they concluded was ...." consider adding a comma before was.

**Complied with.**

Figure 7 would greatly benefit from error bands.

Complied with. CALIOP level 2 products are provided with the estimates of systematic errors in extinction coefficient, back-scattering coefficient and PDR values based on profile

calibration, signal to noise ratios and biases in the lidar ratio assumption. Details of the uncertainty estimation and its propagation in various aerosol parameters are described by Young et al. (2013).

The figure will become complex to infer, and hence two separate figures and associated text have been added in the Supplementary section, Figure S1 for Total extinction coefficient and Figure S2 for Dust extinction coefficient. Both the figures have error bars.

Supplementary material: Pg 2, Ln 1-14:

**Uncertainty in CALIPSO retrievals:**

CALIOP level 2 products are provided with the estimates of systematic errors in extinction coefficient, back-scattering coefficient and PDR values based on profile calibration, signal to noise ratios and biases in the lidar ratio assumption. Details of the uncertainty estimation and its propagation in various aerosol parameters are described in (Young et al., 2013). Present study make use of uncertainty information provided with CALIOP level 2 version 3 data product to screen the data as recommended by (Winker et al., 2013) (Range bins with an uncertainty flag of value greater than or equal to 99.9 km-1 are excluded in the study). Further Cloud screening is carried out with CAD score in the range -70 to -100 to filter the cloud free pixels with high confidence (Liu et al., 2009). To eliminate unstable extinction retrievals, extinction quality control flag (QC) of 0 and 1 are used, that represents the constrained retrievals using transmittance measurements and unconstrained retrievals using stable lidar ratio throughout the iterations, respectively.

---

## Author Response (AR3)

We thank both the reviewers for their constructive comments which have helped us to improve the manuscript. Below are the reviewer comments, in plain italics, and authors replies, in bold.
* * *
**acp-2018-686-RC1**

*The paper reports aircraft measurements of aerosol optical properties collected in three locations in the Indo-Gangetic plain, just prior to the onset of the Indian summer monsoon. The authors report also on the results of radiative forcing and heating rates calculations. They conclude that the knowledge of the vertical distribution of the aerosol optical properties is key to perform accurate calculations. The dataset is surely interesting, and the paper is overall clearly written. I believe this work can be important to the community and should be published after addressing a few issues that I will discuss next.*

**GENERAL COMMENTS**

*The main issue with this analysis is a lack of careful discussion of the uncertainties associated with the different quantities presented. Such uncertainties can affect how the data are interpreted and especially how strong the conclusions might be. Therefore, before publication, the authors should discuss and include an analysis of the uncertainties to account for random as well as systematic errors (for example, the corrections applied to the Aethalometer data are mentioned, but uncertainties associated with those methods and parameters are not included nor discussed).*

**We thank the reviewer for constructive comments. In the revised manuscript we have discussed the uncertainties associated with various parameters, and have interpreted the results in light of those.**

**Uncertainties in $\sigma_{scat}$ :**

**Major uncertainty associated with the scattering measurements is due to non-ideality of the Nephelometer viz. not able to sense the scattering signal for angles $<7^0$ and $> 170^0$ – referred to as truncation error. Scattering measurements were corrected for the truncation error following Anderson and Ogren (1998) and Anderson et al. (1996). They have estimated uncertainties in $\sigma_{scat}$ measurements to be within ~ ±10%.**

**Following lines have been added to the manuscript:**

**Pg 11, Ln14-15: Uncertainties in measured $\sigma_{scat}$ are within ~ ±10% (Anderson et al., 1996).**

**Uncertainties in $\sigma_{ext}$ :**

**Pg 11, Ln 4-6: Massoli et al., (2010) have established that the CAPS PM$_{ex}$ has a detection limit of 3 Mm$^{-1}$ or less at 1 second time resolution and has an uncertainty of ±3%.**

**Uncertainties in σ$_{abs}$ :**

**Pg 12, Ln 4-16: Uncertainties related to measurement of absorption coefficient, using filter based techniques, have been discussed in a series of literatures (Müller et al., 2011;Drinovec et al., 2015;Collaud Coen et al., 2010;Segura et al., 2014;Lack et al., 2014). These uncertainties mainly stem from two major reasons: i) multiple-scattering within the filter fiber matrix, and ii) lower attenuation coefficients for higher filter loadings – filter loading effect (Weingartner et al., 2003). Lack et al. (2014) have estimated an uncertainty of 12 % - 30 % in σ$_{abs}$ measured using filter-based techniques. However, this assumption is on the higher side for the present study for two reasons: i) the new generation Aethalometer (Drinovec et al., 2015) has in place real time compensation of loading effect which earlier was assumed as a constant; and ii) usage of advanced filter tape material which minimizes the effect due to multiple scattering and can be better characterized. After taking into consideration uncertainties introduced due to flow instabilities (Drinovec et al., 2015) an uncertainty of ~ 10% is taken in the absorption coefficient measurements.**

**Uncertainties in α$_{scat/abs}$ :**

**Pg 17, Ln 8-9: Based on the uncertainties in the measurement of σ$_{scat}$ and σ$_{abs}$, as described in section 2.3 above, the estimated uncertainty in α$_{scat/abs}$ ~14%.**

**Uncertainties in SSA :**

**Pg 19, Ln 7-8: 12%-13% uncertainty is estimated in SSA.**

*Some specific cases are discussed in the next section. Although, of minor importance from a scientific point of view, the authors should consider revising the use of the articles "the" and "a" in the manuscript, sometimes misused, but especially, often missing. Also, the use of some preposition and the punctuation could be improved.*

**Complied with. The manuscript has been revised as per the reviewers suggestion and the usage of articles 'the' and 'a' have been appropriately revised.**

**SPECIFIC COMMENTS**

*Page 6, lines 23-34: "Data points at a particular level lying outside two-sigma values of the level average were also removed." The rationale behind this data filtering choice needs to be provided. It is not clear how such data manipulation might bias the averages.*

**Complied with.**

**It was found that due to occasional appearance of clouds aerosol number concentration increased from otherwise stable values. In order to remove such unavoidable incidences from influencing aerosol properties two-sigma criteria was applied. Overall, < ~ 3% of the measurements were screened out due to this criteria. A uniform criterion was applied to screen in all the measurements.**

**Following lines have been added to the manuscript:**

**Pg 6, Ln 23 – Pg 7, Ln 1-3: It was found that due to occasional appearance of clouds aerosol number concentration increased from otherwise stable values. In order to remove such unavoidable incidences from influencing aerosol properties two-sigma criteria was applied wherein data points at a particular level lying outside two-sigma values of the level average were removed. Overall, < ~ 3% of the measurements were screened out due to this criteria.**

*Page 6, lines 25: "....aircraft has...." maybe should be"....aircraft had...."*

**Complied with.**
**Pg 7, Ln 4-5: The measurements were then repeated at the new level after the aircraft had stabilised its attitude.**

*Page 6, lines 25: comma after "This way"?*

**Complied with.**
**Pg 7, Ln5: This way, 20 minutes of useful data was ensured at each level.**

*Page 7, lines 2: "....data was extracted when...." what do the authors mean by "extracted"? Do they mean removed? Or something else? Please clarify.*

**Sorry for the confusion. The sentence is modified as following:**

**Pg 7, Ln 14-15: Near ground, 0 m - 200 m, data represent measurements when aircraft altitude was below 200 m, as confirmed from GPS data.**

*Page 11, lines 18: What wavelength was used and why this specific mass absorption cross-section? Please clarify. Even more importantly, and related to the general comment above, what is the uncertainty associated with the use of this specific mac? There is a wide range of mac values published in the literature, even for a specific given wavelength and even for laboratory generated aerosols; therefore, this certainly introduces a significant uncertainty in the aerosol absorption and SSA, and will propagate into derived forcings and heating rates.*

**Sorry for the confusion here. Aethalometer measures attenuation of light by aerosols deposited on a filter spot. Absorption coefficient is then calculated from the rate of change of attenuation, filter spot area, and volumetric flow rate using equation given below (Weingartner et al., 2003).**

$$\sigma_{abs=\frac{A}{Q}*\frac{\Delta ATN}{\Delta t}}$$

**where A is the filter spot area, Q is the volumetric flow rate, and ΔATN is change in attenuation in time Δt.**

**BC at 880 nm wavelength is calculated from $\sigma_{abs}$ by assuming a mass-absorption cross-section (mac) value of 7.77 m$^2$.g$^{-1}$. This value denotes the mac of freshly emitted BC aerosols freely suspended in the Air. Hence there is no uncertainty involved in $\sigma_{abs}$ due to an assumption of mac. The only uncertainty which will propagate in the calculation of SSA, radiative forcing, and heating rates is the uncertainty in the measurement of absorption coefficient itself. As discussed earlier, estimation of absorption coefficient using filter based optical attenuation technique incurs two major uncertainties: one associated with multiple scattering, and the other with loading effect (Weingartner et al., 2003;Coen et al., 2009). The dual spot Aethalometer (Drinovec et al., 2015) used in the present study compensates in real time for loading effect. Based on a 2-week Klagenfurt campaign data Drinovec et al. (2015) have calculated the value of the parameter C accounting for multiple scattering correction. They concluded that the value of C, obtained as 1.57 for TFE coated glass filter used in AE33, is strongly dependent on filter tape material used.**

**Following changes have been made in the manuscript:**

**Pg 11, Ln 16-21: The Aethalometer measures attenuation of light by aerosols deposited on a filter spot. Absorption coefficient is then calculated from the rate of change of attenuation, filter spot area, and volumetric flow rate using equation given below (Weingartner et al., 2003).**

$$\sigma_{abs=\frac{A}{Q}*\frac{\Delta ATN}{\Delta t}}$$

**where A is the filter spot area, Q is the volumetric flow rate, and ΔATN is change in attenuation in time Δt.**

*Page11, lines 23: The choice of the factor value of 1.57 needs some discussion. In addition, certainly, some rather large uncertainty is associated with this parameter; that should be discussed and should be accounted for while discussing the significance of differences between different layers (these values could easily change even in the same region) and especially when comparing with other results published in the literature that might have used different correction factors, for example.*

**The factor C, with value 1.57 in the present case, represents the enhancement in optical absorption due to filter tape and varies depending on the filter tape material and its packing density. For the Aethalometer model - AE-33 filter tape used is Tetrafluoroethylene (TFE) - coated glass filter (Pallflex 'Fibre-film'- T60A20) and has a C value of 1.57 (Drinovec et al., 2015). Value of C is constant for a given filter type.**

*Page 12, lines 7:"aboard" maybe should be "onboard"?*

**Complied with.**
**Pg 12, Ln 22-24: After each sortie, the measured data were geo-referenced using high time resolution (1s) GPS data, available from a GPS receiver onboard.**

*Page 14, line 2: Even just from the statistical error bars presented in the graph (meaning even ignoring the other potential sources of error discussed above), the increase in absorption coefficient seems hardly significant in a statistical sense.*

**We agree with the reviewer. The sentence has been modified as following:**

**Pg 14, Ln 16-19: Absorption coefficient over BBR, in the east IGP, remains nearly steady with altitude up to around 2 km, above which it marginally increases (unlike at the other two stations), albeit the increase is within the natural variability of lower levels, showing more absorbing aerosols aloft.**

*Table 2: As mentioned, in the table, the authors should add other uncertainties (e.g., instrument accuracies and precisions etc...)*

**Complied with.**
**Table 2 has been updated and now includes instrument/measurement uncertainty.**

**Table 2: Mean ± standard error of column averaged (from near ground to 3000 m) aerosol radiative properties. Instrument/measurement uncertainties are given in the last column.**

| Parameter | Specific regions over the IGP | | | Instrument/ Measurement uncertainty (%) |
|---|---|---|---|---|
| | West (JDR) | Central (VNS) | East (BBR) | |
| $\sigma_{ext}$ (Mm$^{-1}$) | 79 ± 6 | 95 ± 19 | 75 ± 12 | 3 |
| $\sigma_{scat}$ (Mm$^{-1}$) | 63 ± 5 | 69 ± 14 | 58± 6 | 10 |
| $\sigma_{abs}$ (Mm$^{-1}$) | 16 ± 2 | 26 ± 9 | 15 ± 3 | 10 |
| $\alpha_{scat}$ | 0.9 ± 0.2 | 1.7 ± 0.2 | 2.0 ± 0.1 | 14 |
| SSA (integrated) | 0.84 ± 0.03 | 0.73 ± 0.06 | 0.79 ± 0.06 | 13 |

*Page 16, lines 8-11:Specific to the interpretation of the aethalometer Ångsröm exponent, the authors might be interested in the paper by Fialho, P., Freitas, M. C., Barata, F., Vieira, B., Hansen,a. D. a., &Honrath, R. E. (2006). The Aethalometer calibration and determination of iron concentration in dust aerosols. Journal of Aerosol Science, 37(11), 1497–1506.https://doi.org/10.1016/j.jaerosci.2006.03.002\*

**Thanks for bringing to our attention this important literature which certainly will be useful while interpreting absorption Ångsröm exponent in a subsequent work.**

*Page 16, line 21: ".... reveals the gradual transformation of aerosol...." the term "transformation" might be misinterpreted as the aging of an aerosol, but I believe the authors mean something different. Consider using a different term or clarifying.*

**Complied with. The sentence has been modified as:**

**Pg 18, Ln 1: It clearly reveals a gradual change in aerosol type across the IGP during the pre-monsoon.'**

*Page 19, line 25: Is the "reduction" statistically significant? See similar comments above.*

**The reduction is not statistically significant. The sentence has been modified as:**
**Pg 20, Ln 24 – Pg 21, Ln 1-2: Quite opposite to central IGP, East IGP shows reduction in SSA values, albeit within the uncertainty range, with enhanced variability as we move to higher altitudes.'**

*Page 20, line 4: "This clearly points...." The conclusion seems quite reasonable here, but I am not sure the data strongly ("clearly") support this point because of the large standard errors*

*that make several of the changes hardly significant. Maybe using terminology such as "....suggest" or "....data are compatible with...." might be more appropriate.*

**Complied with. The sentence has been modified as:**
**Pg 21, Ln6-9: This suggest that inaccuracies may arise in the estimation of aerosol absorption and atmospheric forcing and heating rates when using a single columnar SSA value (derived from, for example sunphotometer measurements) and brings out the need for region-specific, altitude resolved values of SSA and estimate layer-by-layer forcing and heating rates for more accurate climate impact assessment.'**

*Figure 6 requires error bars.*

**Complied with.**

[Figure]

**Figure 6:** Seasonal variation of columnar (500 m - 3000 m) SSA over different regions of the IGP: west (square), central (triangle), and east (circle). Values in the upper left circle are calculated from Babu et al., 2016.

*Page 24, line13: "This, they concluded was ...." consider adding a comma before was.*

**Complied with.**
**Pg 25, Ln 13: This, they concluded was, due to presence of …**

*Figure 7 would greatly benefit from error bands.*

**Complied with. CALIOP level 2 products are provided with the estimates of systematic errors in extinction coefficient, back-scattering coefficient and PDR values based on profile calibration, signal to noise ratios and biases in the lidar ratio assumption. Details of the**

uncertainty estimation and its propagation in various aerosol parameters are described by Young et al. (2013).

The figure will become complex to infer, and hence two separate figures and associated text have been added in the Supplementary section, Figure S1 for Total extinction coefficient and Figure S2 for Dust extinction coefficient. Both the figures have error bars.

Following lines have been added to the manuscript:

Pg 26, Ln 23 - 24: Uncertainties in the CALIPSO retrievals are discussed in the supplementary material.

Supplementary material:
Pg 2, Ln 1-14:

Uncertainty in CALIPSO retrievals:

CALIOP level 2 products are provided with the estimates of systematic errors in extinction coefficient, back-scattering coefficient and PDR values based on profile calibration, signal to noise ratios and biases in the lidar ratio assumption. Details of the uncertainty estimation and its propagation in various aerosol parameters are described in (Young et al., 2013). Present study make use of uncertainty information provided with CALIOP level 2 version 3 data product to screen the data as recommended by (Winker et al., 2013) (Range bins with an uncertainty flag of value greater than or equal to 99.9 km-1 are excluded in the study). Further Cloud screening is carried out with CAD score in the range -70 to -100 to filter the cloud free pixels with high confidence (Liu et al., 2009). To eliminate unstable extinction retrievals, extinction quality control flag (QC) of 0 and 1 are used, that represents the constrained retrievals using transmittance measurements and unconstrained retrievals using stable lidar ratio throughout the iterations, respectively. Figure S1 and Figure S2 show the CALIPSO derived total extinction profiles and dust extinction profiles over the stations JDR (left), VNS (centre) and BBR (right), respectively. Error bars indicate the standard deviation around the mean.

[Figure]

**Figure S1:** CALIPSO derived total extinction profiles over the stations JDR (left), VNS (centre) and BBR (right). Error bars indicate the standard deviation around the mean.

[Figure]

**Figure S2:** CALIPSO derived dust extinction profiles over the stations JDR (left), VNS (centre) and BBR (right). Error bars indicate the standard deviation around the mean.

*Page 28, line 9: The acronyms TOA and SUR were defined already just a couple of lines above.*
**Complied with. Repeated acronyms have been removed.**

*Figure 8 requires error bars.*
*Page31, line 21-23: Such comparison needs an uncertainty estimate and the discussion needs to be put in the context of the uncertainties. See previous comments.*
*Figure 9 requires error bars.*

**Complied with.**
**Figure 8 has been updated and the error bars represent standard deviation around the mean.**
**Pg 29, Ln 16: Vertical error bars represent standard deviation around the mean.**

[Figure]

**Figure 8:** Aerosol induced short-wave radiative forcing at the top-of-atmosphere (TOA) (filled black bar), surface (filled white bar), and atmosphere (filled gray bar) for the regions of the IGP: west (JDR) - left; centre (VNS) - centre; and east (BBR) - right panel. Error bars represent standard deviation around the mean.

**Following lines have been added to the manuscript:**

**Pg 29, Ln 8-13: McComiskey et al. (2008) have estimated uncertainty in the calculation of $\Delta F$ to be in the range 20% - 80%. The wide range of uncertainties in $\Delta F$ is due to uncertainties associated with measurements of AOD, SSA, g, and surface reflectance. They have assumed 3% uncertainty in SSA calculation. In the present case SSA uncertainties are ~13% hence the uncertainties in $\Delta F$ calculations are likely to be on the higher side**.

**Pg 29, Ln 23 – 25: Significant variability in $(\Delta F)_{ATM}$ values over western and eastern IGP stem from high vertical variation in $\sigma_{ext}$ whereas central IGP has significant variability in $\sigma_{ext}$ only near to surface.**

**Pg 33, Ln 8-15: Uncertainties in the calculation of heating rate stem from uncertainties associated with the measurements of AOD, SSA, surface reflectance, and calculation of g. An uncertainty of 20% - 80% in the calculation of $\Delta F$ is reported by McComiskey et al. (2008). Hence heating rate profiles presented here are likely to have the same uncertainties band as associated with $\Delta F$. Considering the uncertainties the underestimation or overestimation of heating rates may not be statistically significant in the central and the west IGP but is significant at upper levels in eastern IGP.**

*Page 33, line 3: Consider revising the sentence "Analysis of a decadal (2001-2010) aerosol data...." to "Analysis of decadal (2001-2010) aerosol data...." or "Analysis of a decadal (2001-2010) aerosol dataset....". In the same sentence, "AERONET station" is this a specific station? If so, please provide the location. Or are they several "stations"*

**Corrected.**
**Pg 34, Ln 13 – Pg 35, Ln1: Analysis of a decadal (2001-2010) aerosol dataset, from AERONET station in Kanpur,**

*Page 33, line 6: "of" in front of "1.5 km"? Also, consider adding a comma after "above" and before "there".*

**Corrected.**
**Pg 35, Ln 3-4: … due to enhanced absorption by aerosols at altitudes of 1.5 km and above, there is …**

*Page 33, line 8: Are these "2 to 3 C" or "2 to 3 C/d"? Also, consider using consistent units (meaning K instead of C)*

**Complied with. Corrected to '2 to 3 K'.**
**Pg 35, Ln 5: … cooling of about 2 to 3 K in the IGP.**

*Page34, line 3: "....both extrinsic and intrinsic...." what does that mean? Do the authors mean "....both intensive and extensive...." or something else?*

**Sorry for the confusion here. The terminologies 'extrinsic and intrinsic' have been changed to 'intensive and extensive'.**
**Pg 36, Ln 25: … to characterize aerosol radiative properties, both intensive and extensive, and its impact on …**

*Page 34, line 14: The term "significant" should probably be used exclusively with a statistical meaning; so, is this "reduction" indeed significant from a statistical point of view? Or not? See also comments above.*

**Complied with. We agree with the reviewer in the usage of the term 'significant'. The word 'significant' has been dropped.**
**Pg 36, Ln 11: Reduction in SSA values, from …**

*Page 34, line 22: the verb "become" seems to imply a transformation of the aerosol over time, while here I think the different sources and transport patterns might be more dominant?*

**The sentence has been modified as:**

**Pg 36, Ln 19-22: Aerosols across the Indo-Gangetic Plain become highly absorbing prior to onset of the Indian Summer Monsoon, compared to winter and spring, due to change in source strength and transport patterns.**

*Page 35, line 5: see the previous comment on the use of "significantly"*

**Complied with. The word 'significantly' has been replaced with 'remarkably'.**
**Pg 37, Ln 2: … values differed remarkably for highly absorbing/scattering aerosol …**

*Page 35, line 10: "occurs" should be "occur"; also, "...implications to...."*
*or ".......implications for....."?*

[revised manuscript text omitted]

Winker, D. M., Tackett, J. L., Getzewich, B. J., Liu, Z., Vaughan, M. A., and Rogers, R. R.: The global 3-D distribution of tropospheric aerosols as characterized by CALIOP, Atmos. Chem. Phys., 13, 3345-3361, 10.5194/acp-13-3345-2013, 2013.

Young, S. A., Vaughan, M. A., Kuehn, R. E., and Winker, D. M.: The retrieval of profiles of particulate extinction from Cloud-Aerosol Lidar and Infrared Pathfinder Satellite Observations (CALIPSO) data: Uncertainty and error sensitivity analyses, Journal of Atmospheric and Oceanic Technology, 30, 395–428, 10.1175/JTECH-D-12-00046.1, 2013.
*The manuscript describes aerosol optical property measurements using a small aircraft platform during pre-monsoon period over Northern India during 2016. This is likely that first such reported measurements from this part of India and hence quite interesting. In large part these measurements establish previous reports on aerosol property variations going from West to east over this part of India and provides some insights into the variability in altitude.*

**We thank the reviewer for his constructive comments.**

*A key missing component in the discussion is an evaluation of mixing and its affect of the vertical profiles reported here.*

**We didn't had any instrument onboard for measuring mixing state of aerosols.**

*Unfortunately, the aircraft is limited in its ability to reach higher latitudes (non-pressurized) and hence limited to 3000m above ASL. I suspect this may be within the PBL for all the measurements reported here and the profile is primarily established by strong emissions during the sampling time and possibly an artifact of the time the measurements were made. A slightly later time would probably erase this vertical variability. The PBL height at all three locations is probably above 3km during the pre-monsoon time. I would like to see some more discussion of the meteorological/dynamic state of the atmosphere to put the measurements in context.*
*What time of the day where these measurements made? Where all the measurements at all these sites made at the same time of the day? What is the PBL height at the time of measurements? Are there any measurements of temperature profiles that are available close to the regions where the aircraft flew?*

**The reviewer rightly pointed out that the aircraft had altitude limitations (~3000m AMSL) due to flying under un-pressurized mode. However, we will like to stress that the PBL height over the IGP prior to monsoon onset was well below 3 km. PBL heights were obtained, for the flight sortie days, from the NCEP/NCAR global reanalysis product at 0.25\*0.25 deg. grid resolution data. No temperature profiles were available in the nearby regions. Mean PBL height, at local noon time, over the IGP regions for the campaign**

period were: 1.3 ± 0.5 km for JDR (west IGP), 2.3 ± 0.5 km for VNS (central IGP), and 1.4 ± 0.2 km for BBR (east IGP). The observed profiles of aerosol optical properties, prior to monsoon onset, are characteristic of the regions. All the aircraft sorties, at all the sites, were made between ~10:00 - 14:00 hours IST. This was done in order to ensure that the convective boundary layer is evolved, aerosols are well mixed within the column, and there is no residual layer aloft.

Following text has been added to the manuscript:
Pg 7, Ln 6-13: All the aircraft sorties, at all the sites, were made between ~10:00 - 14:00 hours IST. This was done in order to ensure that the convective boundary layer is evolved, aerosols are well mixed within the column, and there is no residual layer aloft. Planetary boundary layer (PBL) heights were obtained, for the flight sortie days, from the NCEP/NCAR global reanalysis product at 0.25*0.25 deg. grid resolution data. Mean PBL height, at local noon time, over the IGP regions for the campaign period were: 1.3 ± 0.5 km for JDR (west IGP), 2.3 ± 0.5 km for VNS (central IGP), and 1.4 ± 0.2 km for BBR (east IGP).

*Manoharan et al., 2013 established that the aerosol absorption in this region has a correlation to the size of the particle. Is there any data on aerosol size that can be shared?*

There was no size cut on absorption measurements. Correlation with particle size cannot be ruled out over this region as rightly pointed by the reviewer but with present measurements will not be possible to discern and quantify.

*Measurements over eastern India indicate less absorbing particles (may be more sulfate?). It will be helpful to have a more extended discussion the scattering at the eastern site and how it may relate to radiative forcing.*

Complied with. Following lines have been added to the manuscript:
Pg 29, Ln 25 – Pg 30, Ln 1-7: Over the eastern IGP scattering coefficient (58 ± 6) is comparable to that over the west (63 ± 5) and central (69 ± 14) IGP. However, absorption coefficient is less as compared to central IGP. Due to significant industrial activities in the eastern IGP, source of sulfate, combined with sea-breeze, source of sea-salt, aerosols over the region are scattering in nature, as indicated by relatively high SSA values of 0.79 as compared to central IGP SSA (0.73). TOA forcing in eastern IGP is -5.4 Wm$^{-2}$ as against -6.5 Wm$^{-2}$ over central IGP. TOA cooling over eastern IGP is less as compared to central IGP, due to their lower abundance.

*I agree with the 'cauldron of complex aerosol type's'. (page 3, line 24) and it may be time to start focusing on the 'other' aerosols in the region besides just absorbing aerosols (sulfates and*

*nitrates). These and SOA may have significant health impacts for the region and deserve more attention.*

[revised manuscript text omitted]

**Uncertainty in CALIPSO retrievals:**

CALIOP level 2 products are provided with the estimates of systematic errors in extinction coefficient, back-scattering coefficient and PDR values based on profile calibration, signal to noise ratios and biases in the lidar ratio assumption. Details of the uncertainty estimation and its propagation in various aerosol parameters are described in (Young et al., 2013). Present study make use of uncertainty information provided with CALIOP level 2 version 3 data product to screen the data as recommended by (Winker et al., 2013) (Range bins with an uncertainty flag of value greater than or equal to 99.9 km-1 are excluded in the study). Further Cloud screening is carried out with CAD score in the range -70 to -100 to filter the cloud free pixels with high confidence (Liu et al., 2009). To eliminate unstable extinction retrievals, extinction quality control flag (QC) of 0 and 1 are used, that represents the constrained retrievals using transmittance measurements and unconstrained retrievals using stable lidar ratio throughout the iterations, respectively. Figure S1 and Figure S2 show the CALIPSO derived total extinction profiles and dust extinction profiles over the stations JDR (left), VNS (centre) and BBR (right), respectively.

[Figure]

**Figure S1:** CALIPSO derived total extinction profiles over the stations JDR (left), VNS (centre) and BBR (right). Error bars indicate the standard deviation around the mean.

[Figure]

**Figure S2:** CALIPSO derived dust extinction profiles over the stations JDR (left), VNS (centre) and BBR (right). Error bars indicate the standard deviation around the mean.

Liu, Z., Vaughan, M., Winker, D., Kittaka, C., Getzewich, B., and Kuehn, R.: The CALIPSO lidar cloud and aerosol discrimination: Version 2 algorithm and initial assessment of performance, Journal of Atmospheric and Oceanic Technology, 26, 1198–1213, 10.1175/2009JTECHA1229.1, 2009.

Winker, D. M., Tackett, J. L., Getzewich, B. J., Liu, Z., Vaughan, M. A., and Rogers, R. R.: The global 3-D distribution of tropospheric aerosols as characterized by CALIOP, Atmos. Chem. Phys., 13, 3345-3361, 10.5194/acp-13-3345-2013, 2013.

Young, S. A., Vaughan, M. A., Kuehn, R. E., and Winker, D. M.: The retrieval of profiles of particulate extinction from Cloud-Aerosol Lidar and Infrared Pathfinder Satellite Observations (CALIPSO) data: Uncertainty and error sensitivity analyses, Journal of Atmospheric and Oceanic Technology, 30, 395–428, 10.1175/JTECH-D-12-00046.1, 2013.